

**OpenArray v1.0: A Simple Operator Library for the Decoupling of**
**Ocean Modelling and Parallel Computing**
Xiaomeng Huang[1,2,3], Xing Huang[1,3], Dong Wang[1,3], Qi Wu[1], Shixun Zhang[3], Yuwen
Chen[1], Mingqing Wang[1,3], Yi Li[3], Yuan Gao[1], Qiang Tang[1], Yue Chen[1], Zheng Fang[1],
Zhenya Song[2,4], Guangwen Yang[1,3]
[1] Ministry of Education Key Laboratory for Earth System Modeling, Department of
Earth System Science, Tsinghua University, Beijing 100084, China
[2] Laboratory for Regional Oceanography and Numerical Modeling, Qingdao National
Laboratory for Marine Science and Technology, Qingdao, 266237, China
[3] National Supercomputing Center in Wuxi, Wuxi, 214011, China
[4] First Institute of Oceanography, Ministry of Natural Resources, Qingdao, 266061,
China
Corresponding author: hxm@tsinghua.edu.cn
**Abstract**
The increasing complexity of climate models combined with rapidly evolving
computational techniques introduces a large gap in climate modelling. In this work, we
design a simple computing library to decouple the work of ocean modelling from the
work of parallel computing. The library provides twelve basic operators that feature
user-friendly interfaces, effective programming and automatic parallelization. We
further implement a highly readable and efficient ocean model that contains only 1860
lines of code but achieves a 91% parallel efficiency in strong scaling and 99% parallel
efficiency in weak scaling with 4096 Intel CPU cores. This ocean model also exhibits
excellent scalability on the Sunway TaihuLight supercomputer. This work presents a
valuable example for the development of the next generation of ocean models.
**Keywords**: automatic parallelization, operator, ocean modelling, parallel computing



## 1. Introduction

Numerous climate models have been developed in the past several decades to improve the predictive understanding of the climate system (Bonan and Doney, 2018; Collins et al., 2018; Taylor et al., 2012). These models are becoming increasingly complicated, and the amount of code has expanded from a few thousand lines to tens of thousands of lines, or even millions of lines. In terms of software engineering, an increase in code causes the models to be more difficult to develop and maintain.

The complexity of these models mainly originates from three aspects. First, more model components and physical processes have been embedded into the climate model, leading to a tenfold increase in the amount of code (Alexander and Easterbrook, 2015). Second, some heterogeneous and advanced computing platforms (Lawrence et al., 2018) have been widely applied by the climate community, resulting in a fivefold increase in the amount of code (Xu et al., 2015). Last, most of the model program needs to be rewritten due to the continual development of novel numerical methods and meshes. The promotion of novel numerical methods and technologies produced in the fields of computational mathematics and computer science have been limited in climate science because of the extremely heavy burden caused by program rewriting and migration.

Over the next few decades, tremendous computing capacities will be accompanied by more heterogeneous architectures, thus making for a much more sophisticated computing environment for climate modellers than ever before (Bretherton et al., 2012). Clearly, transiting the current climate models to the next generation of computing environments will be highly challenging and disruptive. Overall, complex climate model codes combined with rapidly evolving computational techniques create a very large gap in climate science.

To reduce the complexity of climate models and bridge this gap, we believe that a universal and productive computing library is probably the solution. Through



establishing an implicit parallel and platform-independent computing library, the
complex models can be simplified and will no longer need explicit parallelization and
transiting, thus effectively decoupling the development of ocean models from
complicated parallel computing techniques and diverse heterogeneous computing
platforms.

Many studies have addressed the complexity of parallel programming for numerical
simulations. Operator overloading is one of the mainstream implementations and is
fairly straightforward (Corliss and Griewank, 1994; Walther et al., 2003). However, this
method is prone to work inefficiency because overloading execution induces numerous
unnecessary intermediate variables, consuming valuable memory bandwidth resources.
Using a source-to-source translator offers another solution. The important design
philosophy of this method is dependent on the simple self-defined rules in the former
language to automatically generate code conforming to the latter language (Bae et al.,
2013; Lidman et al., 2012). In the MIT General Circulation Model (MITgcm), the
modellers use OpenAD (Naumann et al., 2006; Utke et al., 2008), which is an automatic
algorithmic differentiation tool with a set of mathematical and linguistic rules, to
generate fairly efficient tangent linear and adjoint code (Adcroft et al., 2017). Moreover,
some outstanding domain specific languages (DSL), such as ATMOL (van Engelen,
2001), ICON DSL (Torres et al., 2013) and STELLA (Gysi et al., 2015), provide high-
level abstraction interfaces that use mathematical notations similar to those used by
domain scientists so that they can write much more concise and simpler code.

In fact, when using source-to-source translator and DSL methods to develop practical
climate models, one major difficulty is the requirement of a stable and robust compiler,
rather than an experimental compiler, at the product level. Another difficulty is that the
climate modellers have to change their programming habits and master a new
programming method through novel rules or DSLs instead of using Fortran, which they
are most familiar with. The last difficulty is that although a small part of the existing



source-to-source translators and DSLs currently support graphics processing units
(GPUs), most of the source-to-source translators and DSLs still do not support the
rapidly evolving heterogeneous computing platforms, especially the Chinese Sunway
TaihuLight supercomputer located at the National Supercomputing Center in Wuxi.

Inspired by the philosophy of operator overloading, source-to-source translating and
DSLs, we integrated the advantages of these three methods into a simple computing
library which is called OpenArray. The main contributions of OpenArray are as follows:
• Easy-to-use. The modellers can write simple operator expressions in Fortran to
solve partial differential equations (PDEs). The entire program appears to be
serial and the modellers do not need to know any parallel computing techniques.
We summarized twelve basic generalized operators to support whole model
calculations in ocean models using the finite difference method and staggered
grid in OpenArray.
• High efficiency. We adopt some advanced methods, including intermediate
computation graphing, asynchronous communication, kernel fusion, loop
optimization, and vectorization, to decrease the consumption of memory
bandwidth and improve efficiency. Performance of the programs implemented
by OpenArray is similar to that of original parallel program manually optimized
by experienced programmers.
• Portability. The current OpenArray version support both CPU and Sunway
platforms. The input of OpenArray is a Fortran source file including the operator
expression form; then, the intermediate C++ code is automatically generated by
OpenArray. The final output is a program that is executable on different
computing platforms.

Furthermore, we developed a practical ocean model based on the Princeton Ocean
Model (POM, Blumberg and Mellor, 1987) to test the capability and efficiency of
OpenArray. The new model is called the Generalized Operator Model of the Ocean



(GOMO). Because the parallel computing details are completely hidden, GOMO
consists of only 1860 lines of Fortran code and is more easily understood and
maintained than the original POM. Moreover, GOMO exhibits excellent scalability and
portability to central processing unit (CPU) and Sunway platforms.

The remainder of this paper is organized as follows. Section 2 introduces some concepts
and presents the detailed mathematical descriptions of formulating the PDEs into
operator expressions. Section 3 describes the detailed design and optimization
techniques of OpenArray. Implementation of GOMO is described in section 4. Section
5 evaluates the performance of OpenArray and GOMO. Finally, conclusions are given
in section 6.

**2. Concepts of the Array, Operator, and Abstract Staggered Grid**
In this section, we introduce three important concepts in OpenArray: Array, Operator
and Abstract Staggered Grid to illustrate the design of OpenArray.

**2.1 Array**
To achieve this simplicity, we designed a derived data type, *Array*, which inspired our
project name, OpenArray. The new *Array* data type comprises a series of information,
including a 3-dimensional array to store data, a pointer to the computational grid, a
Message Passing Interface (MPI) communicator, the size of the halo region and other
information about the data distribution. All the information is used to manipulate the 3-
dimensional array as a complete object to simplify the parallel computing. In the
traditional ocean models, calculations for each grid point and the *i, j,* and *k* loops in the
horizontal and vertical directions are unavoidable. The advantage of taking the arrays
as a complete object is the significant reduction in the number of loop operations in the
models, making the code more intuitive and readable. When using OpenArray library
in a program, one can use *type(Array)* to declare new variables.





**2.2 Operator**

To illustrate the concept of an operator, we first take a 2-dimensional (2D) continuous

equation solving sea surface elevation as an example:

$$\frac{\partial \eta}{\partial t} + \frac{\partial DU}{\partial x} + \frac{\partial DV}{\partial y} = 0 \tag{1}$$

where $\eta$ is the surface elevation, $U$ and $V$ are the zonal and meridional velocities, and

$D$ is the depth of the fluid column. We choose the finite difference method and staggered

Arakawa C grid scheme, which are adopted by most regional ocean models. Then, the

above continuous equation can be discretized into the following form.

$$\frac{\eta_{t+1}(i,j) - \eta_{t-1}(i,j)}{2*dt} + \frac{(D(i+1,j)+D(i,j))*U(i+1,j) - (D(i,j)+D(i-1,j))*U(i,j)}{dx(i,j)} +$$

$$\frac{(D(i,j+1)+D(i,j))*V(i,j+1) - (D(i,j)+D(i,j-1))*V(i,j)}{dy(i,j)} = 0 \tag{2}$$

where subscripts $\eta_{t+1}$ and $\eta_{t-1}$ denote the surface elevations at the $(t+1)$ time step and $(t-1)$ time step. To simplify the discrete form, we introduce some notation for the

differentiation ($\delta_f^x$, $\delta_b^y$) and interpolation ($\overline{(\ )}_f^x$, $\overline{(\ )}_b^y$). The $\delta$ and overbar symbols define

the differential operator and average operator. The subscript $x$ or $y$ denotes that the

operation acts in the $x$ or $y$ direction, and the superscript $f$ or $b$ denotes that the

approximation operation is forward or backward.

Table 1 lists the detailed definitions of twelve basic operators. The term *var* denotes a

3-dimenonal model variable. All twelve operators for the finite difference calculations

are named using three letters in the form [A|D][X|Y|Z][F|B]. The first letter contains

two options, A or D, indicating an average or a differential operator. The second letter

contains three options, X, Y or Z, representing the direction of operation. The last letter

contains two options, F or B, representing forward or backward operation. The *dx*, *dy*

and *dz* are the distances between two adjacent grid points along the *x*, *y* and *z* directions.

Using the basic operators, Eq. (2) is expressed as:

$$\frac{\eta_{t+1} - \eta_{t-1}}{2*dt} + \delta_f^x(\overline{D}_b^x * U) + \delta_f^y(\overline{D}_b^y * V) = 0 \tag{3}$$

Thus,

$$\eta_{t+1} = \eta_{t-1} - 2*dt*\left(\delta_f^x(\overline{D}_b^x * U) + \delta_f^y(\overline{D}_b^y * V)\right) \tag{4}$$

Then, Eq. (4) can be easily translated into a line of code using operators (the bottom





left panel in Fig. 1). Compared with the pseudo-codes (the right panel), the
corresponding implementation by operators is simpler and more consistent with the
equations.

Next, we will use the operators in shallow water equations, which are more complicated
than those in the previous case. Assuming that the flow is in hydrostatic balance and
that the density and viscosity coefficient are constant, and neglecting the molecular
friction, the shallow water equations are:
$\frac{\partial \eta}{\partial t} + \frac{\partial DU}{\partial x} + \frac{\partial DV}{\partial y} = 0$                (5)
$\frac{\partial DU}{\partial t} + \frac{\partial DUU}{\partial x} + \frac{\partial DVU}{\partial y} - fVD = -gD\frac{\partial \eta}{\partial x} + \mu D(\frac{\partial^2 U}{\partial x^2} + \frac{\partial^2 U}{\partial y^2})$                (6)
$\frac{\partial DV}{\partial t} + \frac{\partial DUV}{\partial x} + \frac{\partial DVV}{\partial y} + fUD = -gD\frac{\partial \eta}{\partial y} + \mu D(\frac{\partial^2 V}{\partial x^2} + \frac{\partial^2 V}{\partial y^2})$                (7)
where $f$ is the Coriolis parameter, $g$ is the gravitational acceleration, and $\mu$ is the
coefficient of kinematic viscosity. Using the Arakawa C grid and leapfrog time
difference scheme, the discrete forms represented by operators are shown in Eq. (8) ~
Eq. (10).
$\frac{\eta_{t+1} - \eta_{t-1}}{2*dt} + \delta_f^x \left( \overline{D}_b^x * U \right) + \delta_f^y \left( \overline{D}_b^y * V \right) = 0$                (8)
$\frac{D_{t+1}U_{t+1} - D_{t-1}U_{t-1}}{2*dt} + \delta_b^x \left( \overline{\overline{D}_b^x * U}_f^x * \overline{U}_f^x \right) + \delta_f^y \left( \overline{\overline{D}_b^y * V}_b^x * \overline{U}_b^y \right) - \overline{f \, \overline{V}_f^y * D}_b^x = -g *$
$\overline{D}_b^x * \delta_b^x(\eta) + \mu * \overline{D}_b^x * \left( \delta_b^x \left( \delta_f^x (U_{t-1}) \right) + \delta_f^y \left( \delta_b^y (U_{t-1}) \right) \right)$                (9)
$\frac{D_{t+1}V_{t+1} - D_{t-1}V_{t-1}}{2*dt} + \delta_f^x \left( \overline{\overline{D}_b^x * U}_b^y * \overline{V}_b^x \right) + \delta_b^y \left( \overline{\overline{D}_b^y * V}_f^y * \overline{V}_f^y \right) + \overline{f \overline{U}_f^x * D}_b^y = -g *$
$\overline{D}_b^y * \delta_b^y(\eta) + \mu * \overline{D}_b^y * \left( \delta_f^x \left( \delta_b^x (V_{t-1}) \right) + \delta_b^y \left( \delta_f^y (V_{t-1}) \right) \right)$                (10)
As the shallow water equations are solved, spatial average and difference operations
are called repeatedly. Such operations consume the vast majority of the computing
resources when solving the shallow water equations. Therefore, it is necessary to
abstract these common operations from PDEs and encapsulate them into user-friendly,
platform-independent implicit parallel operators. As shown in Fig. 2, we require only 3
lines of code to solve the shallow water equations. This more realistic case suggests

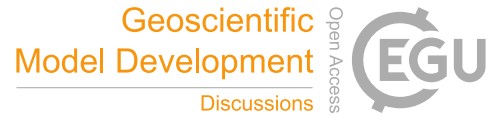



that even more complex PDEs can be constructed and solved by following this elegant
approach.

**2.3 Abstract staggered grid**
Most ocean models are implemented on the basis of the staggered Arakawa grids
(Arakawa and Lamb, 1981; Griffies et al., 2000). The variables in ocean models are
allocated at different grid points. The calculations that use these variables are performed
after several reasonable interpolations or differences. When we call the differential
operations on a staggered grid, the difference value between adjacent points should be
divided by the grid increment to obtain the final result. Setting the correct grid
increment for modellers is troublesome work that is extremely prone to error, especially
when the grid is nonuniform. Therefore, we proposed an abstract staggered grid to
support flexible switching of operator calculations among different staggered grids.
When the grid information is provided at the initialization phase of OpenArray, the
operators can automatically set the correct grid increments for different *Array* variables.

As shown in Fig. 3, the cubes in the (a), (b), (c), and (d) panels are the minimum abstract
grid accounting for 1/8 of the volume of cube in the (e) panel. The eight points of each
cube are numbered sequentially from 0 to 7, and each point has a set of grid increments,
i.e., *dx*, *dy* and *dz*. For example, all the variables of an abstract Arakawa A grid are
located at Point 3. For the Arakawa B grid, the horizontal velocity *Array* (*U, V*) are
located at Point 0, the temperature (*T*), the salinity (*S*), and the depth (*D*) are located at
Point 3, and the vertical velocity *Array* (*W)* is located at Point 7. For the Arakawa C
grid, *Array U* is located at Point 2 and *Array V* is located at Point 1. In contrast, for the
Arakawa D grid, *Array U* is located at Point 1 and *Array V* is located at Point 2.

When we call the average and differential operators mentioned in Table 1, for example,
on the abstract Arakawa C grid, the position of *Array D* is Point 3, and the average *AXB*
operator acting on *Array D* will change the position from Point 3 to Point 1. Since *Array*

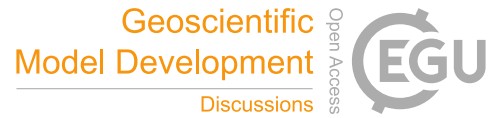



*U* is also allocated at Point 1, the operation *AXB(D)\*U* is allowed. In addition, the
subsequent differential operator on *Array AXB(D)\*U* will change the position of *Array*
*DXF(AXB(D)\*U)* from Point 1 to Point 3.

The jumping rules of different operators are given in Table 2. Due to the design of the
abstract staggered grids, the jumping rules for the Arakawa A, B, C, and D grids are
fixed. A change in the position of an array is determined only by the direction of a
certain operator acting on that array.

The position information and jumping rules can be used to automatically check whether
the discrete form of an equation is correct. The grid increments are hidden by all the
differential operators, making the code simple and clean. In addition, since the rules are
suitable for multiple staggered Arakawa grids, the modellers can flexibly switch the
ocean model between different Arakawa grids. Notably, the users of OpenArray should
input the correct positions of each array in the initialization phase. The value of the
position is an input parameter when declaring an *Array*. An error will be reported if an
operation is performed between misplaced points.

Although most of the existing ocean models use finite difference or finite volume
methods on structured or semi-structured meshes, such as POM, the Modular Ocean
Model (MOM) (Griffies, 2012), the Parallel Ocean Program (POP) (Smith et al., 2010),
MITgcm (Adcroft et al., 2017), and the Regional Ocean Modeling System (ROMS)
(Shchepetkin and McWilliams, 2005), there are still some ocean models using
unstructured meshes, including Advanced Circulation model (ADCIRC) (Luettich et
al., 1992), Finite-Volume Coastal Ocean Model (FVCOM) (Chen et al., 2003), and
Stanford Unstructured Nonhydrostatic Terrain-following Adaptive Navier-Stokes
Simulator (SUNTANS) (Fringer et al., 2006), and even the spectral element method
(e.g. Levin et al., 2000). In our current work, we design the basic operator only for finite
different and finite volume methods with structured grids. More customized operator





for the other numerical methods and meshes will be implemented in our future work.

## 3. Design of OpenArray

Through the above operator notations in Table 1, ocean modellers can quickly convert
the discrete PDE equations into the corresponding operator expression forms. The main
purpose of OpenArray is to make complex parallel programming transparent to the
modellers. As illustrated in Fig. 4, we use a computation graph as an intermediate
representation, meaning that the operator expression forms written in Fortran will be
translated into a computation graph with a particular data structure. In addition,
OpenArray will use the intermediate computation graph to analyse the dependency of
the distributed data and automatically produce the underlying parallel code. Finally, we
use stable and mature compilers, such as the GNU Compiler Collection (GCC), Intel
compiler (ICC), and Sunway compiler (SWACC), to generate the executable program
according to different backend platforms. These four steps and some related techniques
are described in detail in this section.

### 3.1 Operator expression

Although the basic generalized operators listed in Table 1 are only suitable to execute
first-order difference, other high-order difference or even more complicated operations
can be combined by these basic operators. For example, a second-order difference
operation can be expressed as $\delta_f^x(\delta_b^x(var))$. Supposing the grid distance is uniform,
the corresponding discrete form is [*var(i+1,j,k)+var(i-1,j,k) -2\* var(i,j,k)* ] / *dx²*. In
addition, the central difference operation can be expressed as $(\delta_f^x(var) + \delta_b^x(var))/2$
since the corresponding discrete form is [*var(i+1,j,k)-var(i-1,j,k)] / 2dx*.

Using these operators to express the discrete PDE equation, the code and formula are
very similar. We call this effect "the self-documenting code is the formula". Fig. 5
shows the one-to-one correspondence of each item in the code and the items in the sea
surface elevation equation. The code is very easy to program and understand. Clearly,





the basic operators and the combined operators greatly simplify the development and
maintenance of ocean models. The complicated parallel and optimization techniques
will be concealed by these operators. Modellers no longer need to care about details
and escape from the "parallelism swamp", thus they can concentrate on the scientific
issues.

**3.2 Intermediate computation graph**
Considering the example mentioned in Fig. 5, if one needs to compute the term
*DXF(AXB(D)\*u)* with the traditional operator overloading method, one first computes
*AXB(D)* and stores the result into a temporary array (named *tmp1)*, and then executes
(*tmp1\*u*) and stores the result into a new array, *tmp2*. The last step is to compute
*DXF(tmp2)* and store the result in a new array, *tmp3*. Numerous temporary arrays
consume a considerable amount of memory, making the efficiency of operator
overloading is poor.

To solve this problem, we convert an operator expression form into a directed and
acyclic graph, which consists of basic data and function nodes, to implement a lazy
expression evaluation (Bloss et al., 1988; Reynolds, 1999). Unlike the traditional
operator overloading method, we overload all arithmetic functions to generate an
intermediate computation graph rather than to obtain the result of each function. This
method is widely used in deep learning frameworks, e.g., TensorFlow (Abadi et al.,
2016) and Theano (Bastien et al., 2012), to improve computing efficiency. Figure 6
shows the procedure of parsing the operator expression form of the sea level elevation
equation into a computation graph. The input variables in the square boxes include the
sea surface elevation (*elb*), the zonal velocity (*u*), the meridional velocity (*v*) and the
depth (*D*). *dt2* is a constant equal to *2\*dt*. The final output is the sea surface elevation
at the next time step (*elf*). The operators in the round boxes have been overloaded in
OpenArray. In summary, all the operators provided by OpenArray are functions for the
*Array* calculation, in which the "=" notation is the assignment function, the "-" notation
is the subtraction function, the "\*" notation is the multiplication function, the "+"





notation is the addition function, DXF and DYF are the differential functions, and AXF
and AYF are the interpolated functions.

**3.3 Automatic code generation**
Given a computation graph, we design a lightweight engine to automatically generate
the corresponding source code automatically (Fig. 7). Each operator node in the
computation graph is called a kernel. The sequence of all kernels in a graph is usually
fused into a large kernel function. Therefore, the underlying engine schedules and
executes the fused kernel once and obtains the final result directly without any auxiliary
or temporary variables. Simultaneously, the scheduling overhead of the computation
graph and the startup overhead of the basic kernels can be reduced.

Most of the scientific computational applications are limited by the memory bandwidth
and cannot fully exploit the computing power of a processor. Fortunately, kernel fusion
is an effective optimization method to improve memory locality. When two kernels
need to process some data, their fusion holds shared data in the memory. Prior to the
kernel fusion, the computation graph is automatically analysed to find the operator
nodes that can be fused, and the analysis results are stored in several subgraphs. After
being given a series of subgraphs, the underlying engine dynamically generates the
corresponding kernel function in C++ using just-in-time (JIT) compilation techniques
(Suganuma and Yasue, 2005). Notably, the time to compile a single kernel function is
short, but practical applications usually need to be run for thousands of time steps, and
the overhead of generating and compiling the kernel functions for the computation
graph is extremely high. Therefore, we generate a fusion kernel function only once for
each subgraph, and put it into a function pool. Later, when facing the same computation
subgraph, we fetch the corresponding fusion kernel function directly from the pool.

Since the arrays in OpenArray are distributed among different processing units, and the
operator needs to use the data in the neighbouring points, in order to ensure the





correctness, it is necessary to check the data consistency before fusion. The use of
different data splitting methods for distributed arrays can greatly affect computing
performance. The current data splitting method in OpenArray is the widely used block-
based strategy. Solving PDEs on structured grids often divides the simulated domain
into blocks that are distributed to different processing units. However, the difference
and average operators always require their neighbouring points to perform array
computations. Clearly, controlling the communication of the boundary region is tedious
work for ocean modellers.

Therefore, we implemented a general boundary management module to automatically
maintain and update the boundary information so that the modellers no longer need to
address the message communication. The boundary management module uses
asynchronous communication to update and maintain the data of the boundary region,
which is useful for simultaneous computing and communication. These procedures of
asynchronous communication are implicitly invoked when calling the basic kernel or
the fused kernel to ensure that the parallel details are completely transparent to the
modellers.

**3.4 Portable program for different backend platforms**
With dynamic code generation and JIT compilation technology, OpenArray can be
easily migrated to different backend platforms. Currently, we have designed the
corresponding source code generation module for Intel CPU and Sunway processors in
OpenArray.

The Sunway TaihuLight is the third fastest supercomputer in the world, with a
LINPACK benchmark rating of 93 Petaflops provided by a multi-core Sunway
processor that includes 4 core-groups, each of which consists of 64 computing
processing elements (CPEs) and a management processing element (MPE) (Qiao et al.,
2017). To make the most of the computing resources of the Sunway TaihuLight, we

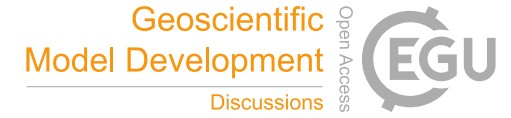



generate kernel functions for the MPE, which is responsible for the thread control, and
CPE, which performs the computations. The kernel functions are fully optimized with
several code optimization techniques (Pugh, 1991) such as loop tiling, loop aligning,
single-instruction multiple-date (SIMD) vectorization, and function inline. In addition,
due to the high memory access latency of CPEs, we accelerate data access by providing
instructions for direct memory access in the kernel to transfer data between the main
memory and local memory (Fu et al., 2017).

**4. Implementation of GOMO**
In this section, we introduce how to implement a practical ocean model using
OpenArray. The most important step is to derive the primitive discrete governing
equations in operator expression form, then the following work will be completed by
OpenArray.

The fundamental equations of GOMO are derived from POM. GOMO features a
bottom-following, free-surface, staggered Arakawa C grid. To effectively evolve the
rapid surface fluctuations, GOMO uses the mode-splitting algorithm to address the fast
propagating surface gravity waves and slow propagating internal waves in barotropic
(external) and baroclinic (internal) modes, respectively. The details of the continuous
governing equations, the corresponding operator expression form and the descriptions
of all the variables used in GOMO are listed in the Appendix A, Appendix B, and
Appendix C, respectively.

Figure 8 shows the basic flow diagram of GOMO. At the beginning of the workflow,
we initialize OpenArray to make all operators suitable for GOMO. After loading the
initial values and the model parameters, the distance information is input into the
differential operators through grid binding. In the external mode, the main consumption
is computing the 2-dimensional sea surface elevation $\eta$ and column-averaged velocity
($Ua$, $Va$). In the internal mode, 3-dimensional array computations predominate in order




to calculate baroclinic motions ($U$, $V$, $W$), tracers ($T$, $S$, $\rho$), and turbulence closure sub-
model ($q^2$, $q^2l$) (Mellor and Yamada, 1982), where ($U$, $V$, $W$) are the velocity fields in
the $x$, $y$ and $\sigma$ directions, ($T$, $S$, $\rho$) are the potential temperature, the salinity and the
density. ($q^2/2$, $q^2l/2$) are the turbulence kinetic energy and production of turbulence
kinetic energy with turbulence length scale.

Because the complicated parallel optimization and tuning processes are decoupled from
the ocean modelling, we completely implemented GOMO based on OpenArray in only
4 weeks, whereas implementation may take several months or even longer when using
the MPI or CUDA library.

In comparison with the existing POM and its multiple variations, to name a few, Stony
Brook Parallel Ocean Model (sbPOM), mpiPOM and POMgpu, GOMO has less code
but is more powerful in terms of compatibility. As shown in Table 3, the serial version
of POM (POM2k) contains 3521 lines of code. sbPOM and mpiPOM are parallelized
using MPI, while POMgpu is based on MPI and CUDA-C. The codes of sbPOM,
mpiPOM and POMgpu are extended to 4801, 9680 and 30443 lines. In contrast, the
code of GOMO is decreased to 1860 lines. Moreover, GOMO completes the same
function as the other approaches while using the least amount of code (Table 4).

In addition, poor portability considerably restricts the use of advanced hardware in
oceanography. With the advantages of OpenArray, GOMO is adaptable to different
hardware architectures, such as the Sunway processor. The modellers do not need to
modify any code when changing platforms, completely eliminating the heavy burden
of transmitting code. As computing platforms become increasingly diverse and
complex, GOMO becomes more powerful and attractive than the machine-dependent
models.



**5、Experimental results**

In this section, we first evaluate the basic performance of OpenArray using benchmark tests on a single CPU platform. After checking the correctness of GOMO through an ideal seamount test case, we use GOMO to further test the scalability and efficiency of OpenArray.

**5.1 Benchmark testing**

We choose four typical PDEs and their implementations from Rodinia v3.1, which is a benchmark suite for heterogeneous computing (Che et al., 2009), as the original version. For comparison, we re-implement these four PDEs using OpenArray. As shown in Table 5, the 2D continuity equation is used to solve sea surface height, and its continuous form is shown in Eq. (1). The 2D heat diffusion equation is a parabolic PDE that describes the distribution of heat over time in a given region. Hotspot is a thermal simulation used for estimating processor temperature on structured grids (Che et al., 2009; Huang et al., 2006). We tested one 2-dimensional case (Hotspot2D) and one 3-dimensional case (Hotspot3D) of this program. The average runtime for 100 iterations is taken as the performance metric. All tests are executed on a single workstation with an Intel Xeon E5-2650 CPU. The experimental results show that the performance of OpenArray versions is comparable to the original versions.

**5.2 Validation tests of GOMO**

The seamount problem proposed by Beckman and Haidvogel is a widely used ideal test case for regional ocean models (Beckmann and Haidvogel, 1993). It is a stratified Taylor column problem, which simulates the flow over an isolated seamount with a constant salinity and a reference vertical temperature stratification. An eastward horizontal current of 0.1 m/s is added at model initialization. The southern and northern boundaries are closed. If the Rossby number is small, an obvious anticyclonic circulation is trapped by the mount in the deep water.



Using the seamount test case, we compare GOMO and sbPOM results. The
configurations of both models are exactly the same. Figure 9 shows that GOMO and
sbPOM both capture the anticyclonic circulation at 3500 metres depth. The shaded plot
shows the surface elevation, and the array plot shows the current at 3500 metres. Figure
9(a), 9(b), and 9(c) are the results of GOMO, sbPOM, and the difference (GOMO-
sbPOM), respectively. The differences in the surface elevation and deep currents
between the two models are negligible (Fig. 9(c)).

**5.3 The weak and strong scalability of GOMO**
The seamount test case is used to compare the performance of sbPOM and GOMO in
a parallel environment. Figure 10(a) shows the result of a strong scaling evaluation, in
which the model size is fixed at 2048×2048×50. The dashed line indicates the ideal
speedup. For the largest parallelisms with 4096 processes, GOMO and sbPOM achieve
91% and 92% parallel efficiency, respectively. Figure 10(b) shows the weak scalability
of sbPOM and GOMO. In the weak scaling test, the model size for each process is fixed
at 128×128×50, and the number of processes is gradually increased from 16 to 4096.
Taking the performance of 16 processes as a baseline, we determine that the parallel
efficiencies of GOMO and sbPOM using 4096 processes are 99.0% and 99.2%,
respectively.

**5.4 Testing on the Sunway platform**
The strong scalability of GOMO is also tested on the Sunway TaihuLight
supercomputer. Supposing that the baseline is the runtime of GOMO at 10000 cores
with a grid size of 4096×4096×50, the parallel efficiency of GOMO can still reach 85%
at 150000 cores, as shown in Fig. 11. However, we notice that the scalability declines
sharply when the number of cores exceeds 150000. There are two reasons leading to
this decline. First, the block size assigned to each core decreases as the number of cores
increases, causing more communication during boundary region updating. Second,
some processes cannot be accelerated even though more computing resources are



available; for example, the time spent on creating the computation graph, generating
the fusion kernels, and compiling the JIT cannot be reduced. In a sense, OpenArray
performs better when processing large-scale data, and GOMO is more suitable for high-
resolution scenarios. In the future, we will further optimize the communication and
graph-creating modules to improve the efficiency for large-scale cores.

**6. Conclusion**
We designed a simple computing library (OpenArray) to decouple ocean modelling and
parallel computing. OpenArray provides twelve basic operators that are abstracted from
PDEs and extended to ocean model governing equations. These operators feature user-
friendly interfaces and an implicit parallelization ability. Meanwhile, some state-of-art
optimization mechanisms, including computation graphing, kernel fusion, dynamic
source code generation and JIT compiling, are applied to boost the performance. The
experimental results prove that the performance of a program using OpenArray is
comparable to that of well-designed programs using Fortran. Based on OpenArray, we
implement a practical ocean model (GOMO) with a high productivity, an enhanced
readability and an excellent scalable performance. Moreover, GOMO shows high
scalability on the Sunway platform. Although more realistic tests are
needed, OpenArray may signal the beginning of a new frontier in future ocean
modelling through ingesting basic operators and cutting-edge computing techniques.

*Code availability.* OpenArray v1.0 is available at
https://github.com/hxmhuang/OpenArray_CXX. GOMO is available at
https://github.com/hxmhuang/GOMO.

**Appendix A: Continuous governing equations**
The equations governing the baroclinic (internal) mode in GOMO are the 3-
dimensional hydrostatic primitive equations.



$\qquad \frac{\partial \eta}{\partial t} + \frac{\partial UD}{\partial x} + \frac{\partial VD}{\partial y} + \frac{\partial W}{\partial \sigma} = 0$ (A1)
$\qquad \frac{\partial UD}{\partial t} + \frac{\partial U^2 D}{\partial x} + \frac{\partial UVD}{\partial y} + \frac{\partial UW}{\partial \sigma} - fVD + gD\frac{\partial \eta}{\partial x} = \frac{\partial}{\partial \sigma}\left(\frac{K_M}{D}\frac{\partial U}{\partial \sigma}\right) +$
$\frac{gD^2}{\rho_0}\frac{\partial}{\partial x}\int_\sigma^0 \rho d\sigma' - \frac{gD}{\rho_0}\frac{\partial D}{\partial x}\int_\sigma^0 \sigma'\frac{\partial \rho}{\partial \sigma'}d\sigma' + F_u$ (A2)
$\qquad \frac{\partial VD}{\partial t} + \frac{\partial UVD}{\partial x} + \frac{\partial V^2 D}{\partial y} + \frac{\partial VW}{\partial \sigma} + fUD + gD\frac{\partial \eta}{\partial y} = \frac{\partial}{\partial \sigma}\left(\frac{K_M}{D}\frac{\partial V}{\partial \sigma}\right) +$
$\frac{gD^2}{\rho_0}\frac{\partial}{\partial y}\int_\sigma^0 \rho d\sigma' - \frac{gD}{\rho_0}\frac{\partial D}{\partial y}\int_\sigma^0 \sigma'\frac{\partial \rho}{\partial \sigma'}d\sigma' + F_v$ (A3)
$\qquad \frac{\partial TD}{\partial t} + \frac{\partial TUD}{\partial x} + \frac{\partial TVD}{\partial y} + \frac{\partial TW}{\partial \sigma} = \frac{\partial}{\partial \sigma}\left(K_H\frac{\partial T}{\partial \sigma}\right) + F_T + \frac{\partial R}{\partial \sigma}$ (A4)
$\qquad \frac{\partial SD}{\partial t} + \frac{\partial SUD}{\partial x} + \frac{\partial SVD}{\partial y} + \frac{\partial SW}{\partial \sigma} = \frac{\partial}{\partial \sigma}\left(K_H\frac{\partial S}{\partial \sigma}\right) + F_S$ (A5)
$\qquad \rho = \rho(T, S, p)$ (A6)
$\qquad \frac{\partial q^2 D}{\partial t} + \frac{\partial Uq^2 D}{\partial x} + \frac{\partial Vq^2 D}{\partial y} + \frac{\partial Wq^2}{\partial \sigma} = \frac{\partial}{\partial \sigma}\left(\frac{K_q}{D}\frac{\partial q^2}{\partial \sigma}\right) + \frac{2K_M}{D}\left[\left(\frac{\partial U}{\partial \sigma}\right)^2 + \left(\frac{\partial V}{\partial \sigma}\right)^2\right] +$
$\frac{2g}{\rho_0}K_H\frac{\partial \rho}{\partial \sigma} - \frac{2Dq^3}{B_1 l} + F_{q^2}$ (A7)
$\qquad \frac{\partial q^2 lD}{\partial t} + \frac{\partial Uq^2 lD}{\partial x} + \frac{\partial Vq^2 lD}{\partial y} + \frac{\partial Wq^2 l}{\partial \sigma} = \frac{\partial}{\partial \sigma}\left(\frac{K_q}{D}\frac{\partial q^2 l}{\partial \sigma}\right) + E_1 l\left\{\frac{K_M}{D}\left[\left(\frac{\partial U}{\partial \sigma}\right)^2 +\right.\right.$
$\left.\left.\left(\frac{\partial V}{\partial \sigma}\right)^2\right] + \frac{gE_3}{\rho_0}K_H\frac{\partial \rho}{\partial \sigma}\right\}\widetilde{W} - \frac{Dq^3}{B_1} + F_{q^2 l}$ (A8)

Where $F_u$, $F_v$, $F_{q^2}$, and $F_{q^2 l}$ are horizontal kinematic viscosity terms of u, v, $q^2$, and
$q^2 l$, respectivly. $F_T$ and $F_S$ are horizontal diffusion terms of T and S respectivly. $\widetilde{W}$
is the wall proximity function.
$\qquad F_u = \frac{\partial}{\partial x}\left(2A_M D\frac{\partial U}{\partial x}\right) + \frac{\partial}{\partial y}\left[A_M D\left(\frac{\partial U}{\partial y} + \frac{\partial V}{\partial x}\right)\right]$ (A9)
$\qquad F_v = \frac{\partial}{\partial y}\left(2A_M D\frac{\partial V}{\partial y}\right) + \frac{\partial}{\partial x}\left[A_M D\left(\frac{\partial U}{\partial y} + \frac{\partial V}{\partial x}\right)\right]$ (A10)
$\qquad F_T = \frac{\partial}{\partial x}\left(A_H H\frac{\partial T}{\partial x}\right) + \frac{\partial}{\partial y}\left(A_H H\frac{\partial T}{\partial y}\right)$ (A11)
$\qquad F_S = \frac{\partial}{\partial x}\left(A_H H\frac{\partial S}{\partial x}\right) + \frac{\partial}{\partial y}\left(A_H H\frac{\partial S}{\partial y}\right)$ (A12)
$\qquad F_{q^2} = \frac{\partial}{\partial x}\left(A_M H\frac{\partial q^2}{\partial x}\right) + \frac{\partial}{\partial y}\left(A_M H\frac{\partial q^2}{\partial y}\right)$ (A13)
$\qquad F_{q^2 l} = \frac{\partial}{\partial x}\left(A_M H\frac{\partial q^2 l}{\partial x}\right) + \frac{\partial}{\partial y}\left(A_M H\frac{\partial q^2 l}{\partial y}\right)$ (A14)
$\qquad \widetilde{W} = 1 + \frac{E_2 l}{\kappa}\left(\frac{1}{\eta - z} + \frac{1}{H - z}\right)$ (A15)

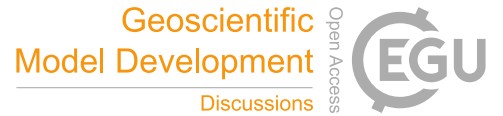



The equations governing the barotropic (external) mode in GOMO are obtained by
vertically integrating the baroclinic equations.
$$\frac{\partial \eta}{\partial t} + \frac{\partial U_A D}{\partial x} + \frac{\partial V_A D}{\partial y} = 0 \qquad (A16)$$

$$\frac{\partial U_A D}{\partial t} + \frac{\partial (U_A)^2 D}{\partial x} + \frac{\partial U_A V_A D}{\partial y} - f V_A D + g D \frac{\partial \eta}{\partial x} = \tilde{F}_{u_a} - wu(0) +$$

$$wu(-1) - \frac{gD}{\rho_0} \int_{-1}^{0} \int_{\sigma}^{0} \left[ D \frac{\partial \rho}{\partial x} - \frac{\partial D}{\partial x} \sigma' \frac{\partial \rho}{\partial \sigma} \right] d\sigma' d\sigma + G_{u_a} \qquad (A17)$$
$$\frac{\partial V_A D}{\partial t} + \frac{\partial U_A V_A D}{\partial y} + \frac{\partial (V_A)^2 D}{\partial y} + f U_A D + g D \frac{\partial \eta}{\partial y} = \tilde{F}_{v_a} - wv(0) +$$

$$wv(-1) - \frac{gD}{\rho_0} \int_{-1}^{0} \int_{\sigma}^{0} \left[ D \frac{\partial \rho}{\partial y} - \frac{\partial D}{\partial y} \sigma' \frac{\partial \rho}{\partial \sigma} \right] d\sigma' d\sigma + G_{v_a} \qquad (A18)$$

Where $\tilde{F}_{u_a}$ and $\tilde{F}_{v_a}$ are the horizontal kinematic viscosity terms of $U_A$ and $V_A$
respectivly. $G_{u_a}$ and $G_{v_a}$ are the dispersion terms of $U_A$ and $V_A$ respectivly. The
subscript 'A' denotes vertical integration.

$$\tilde{F}_{u_a} = \frac{\partial}{\partial x}\left[ 2H(AA_M)\frac{\partial U_A}{\partial x} \right] + \frac{\partial}{\partial y}\left[ H(AA_M)\left( \frac{\partial U_A}{\partial y} + \frac{\partial V_A}{\partial x} \right) \right] \qquad (A19)$$

$$\tilde{F}_{v_a} = \frac{\partial}{\partial y}\left[ 2H(AA_M)\frac{\partial V_A}{\partial y} \right] + \frac{\partial}{\partial x}\left[ H(AA_M)\left( \frac{\partial U_A}{\partial y} + \frac{\partial V_A}{\partial x} \right) \right] \qquad (A20)$$

$$G_{u_a} = \frac{\partial^2 (U_A)^2 D}{\partial x^2} + \frac{\partial^2 U_A V_A D}{\partial x \partial y} - \tilde{F}_{u_a} - \frac{\partial^2 (U^2)_A D}{\partial x^2} - \frac{\partial^2 (UV)_A D}{\partial y^2} + (F_u)_A \quad (A21)$$

$$G_{v_a} = \frac{\partial^2 U_A V_A D}{\partial x \partial y} + \frac{\partial^2 (V_A)^2 D}{\partial y^2} - \tilde{F}_{v_a} - \frac{\partial^2 (UV)_A D}{\partial x^2} - \frac{\partial^2 (V^2)_A D}{\partial y^2} + (F_v)_A \quad (A22)$$

$$U_A = \int_{-1}^{0} U d\sigma \qquad (A23)$$

$$V_A = \int_{-1}^{0} V d\sigma \qquad (A24)$$

$$(U^2)_A = \int_{-1}^{0} U^2 d\sigma \qquad (A25)$$

$$(UV)_A = \int_{-1}^{0} UV d\sigma \qquad (A26)$$

$$(V^2)_A = \int_{-1}^{0} V^2 d\sigma \qquad (A27)$$

$$(F_u)_A = \int_{-1}^{0} F_u d\sigma \qquad (A28)$$

$$(F_v)_A = \int_{-1}^{0} F_v d\sigma \qquad (A29)$$





$$AA_M = \int_{-1}^0 (A_M)d\sigma \qquad (A30)$$


**Appendix B: Discrete governing equations**

The discrete governing equations of baroclinic (internal) mode expressed by operators

are shown as below:

$$\frac{\eta^{t+1}-\eta^{t-1}}{2dti} + \delta_f^x(\overline{D_b}^x U) + \delta_f^y(\overline{D_b}^y V) + \delta_f^z(W) = 0 \qquad (B1)$$

$$\frac{(\overline{D_b}^x U)^{t+1}-(\overline{D_b}^x U)^{t-1}}{2dti} + \delta_b^x\left[\overline{(\overline{D_b}^x U)_f}^x \overline{U_f}^x\right] + \delta_f^y\left[\overline{(\overline{D_b}^y V)_b}^x \overline{U_b}^y\right] +$$

$$\delta_f^z(\overline{W_b}^x \overline{U_b}^z) - \overline{(\tilde{f}\overline{V_f}^y D)_b}^x - \overline{(f\overline{V_f}^y D)_b}^x + g\overline{D_b}^x \delta_b^x(\eta) = \delta_b^z\left[\frac{\overline{K_{M_b}}^x}{(\overline{D_b}^x)^{t+1}}\delta_f^z(U^{t+1})\right] +$$

$$\frac{g(\overline{D_b}^x)^2}{\rho_0}\int_\sigma^0\left[\delta_b^x(\overline{\rho_b}^z) - \frac{\sigma}{\overline{D_b}^x}\delta_b^z(\overline{\rho_b}^x)\right]d\sigma' + F_u \qquad (B2)$$

$$\frac{(\overline{D_b}^y V)^{t+1}-(\overline{D_b}^y V)^{t-1}}{2dti} + \delta_f^x\left[\overline{(\overline{D_b}^x U)_b}^y \overline{V_b}^x\right] + \delta_b^y\left[\overline{(\overline{D_b}^y V)_f}^y \overline{V_f}^y\right] +$$

$$\delta_f^z(\overline{W_b}^y \overline{V_b}^z) + \overline{(\tilde{f}\overline{U_f}^x D)_b}^y + \overline{(f\overline{U_f}^x D)_b}^y + g\overline{D_b}^y \delta_b^y(\eta) = \delta_b^z\left[\frac{\overline{K_{M_b}}^y}{(\overline{D_b}^y)^{t+1}}\delta_f^z(V^{t+1})\right] +$$

$$\frac{g(\overline{D_b}^y)^2}{\rho_0}\int_\sigma^0\left[\delta_b^y(\overline{\rho_b}^z) - \frac{\sigma}{\overline{D_b}^y}\delta_b^z(\overline{\rho_b}^y)\right]d\sigma' + F_v \qquad (B3)$$

$$\frac{(TD)^{t+1}-(TD)^{t-1}}{2dti} + \delta_f^x(\overline{T_b}^x U\overline{D_b}^x) + \delta_f^y(\overline{T_b}^y V\overline{D_b}^y) + \delta_f^z(\overline{T_b}^z W) =$$

$$\delta_b^z\left[\frac{K_H}{D^{t+1}}\delta_f^z(T^{t+1})\right] + F_T + \delta_f^z R \qquad (B4)$$

$$\frac{(SD)^{t+1}-(SD)^{t-1}}{2dti} + \delta_f^x(\overline{S_b}^x U\overline{D_b}^x) + \delta_f^y(\overline{S_b}^y V\overline{D_b}^y) + \delta_f^z(\overline{S_b}^z W) =$$

$$\delta_b^z\left[\frac{K_H}{D^{t+1}}\delta_f^z(S^{t+1})\right] + F_S \qquad (B5)$$

$$\rho = \rho(T,S,p) \qquad (B6)$$

$$\frac{(q^2 D)^{t+1}-(q^2 D)^{t-1}}{2dti} + \delta_f^x(\overline{U_b}^z \overline{q^2}_b^x \overline{D_b}^x) + \delta_f^y(\overline{V_b}^z \overline{q^2}_b^y \overline{D_b}^y) +$$

$$\delta_f^z\overline{(Wq^2)}_b^z = \delta_b^z\left[\frac{\overline{K_{q_f}}^z}{D^{t+1}}\delta_f^z(q^2)^{t+1}\right] + \frac{2K_M}{D}\left\{\left[\delta_b^z(\overline{U_f}^x)\right]^2 + \left[\delta_b^z(\overline{V_f}^y)\right]^2\right\} +$$

$$\frac{2g}{\rho_0}K_H\delta_b^z(\rho) - \frac{2Dq^3}{B_1 l} + F_{q^2} \qquad (B7)$$

$$\frac{(q^2 lD)^{t+1}-(q^2 lD)^{t-1}}{2dti} + \delta_f^x(\overline{U_b}^z \overline{q^2 l_b}^x \overline{D_b}^x) + \delta_f^y(\overline{V_b}^z \overline{q^2 l_b}^y \overline{D_b}^y) +$$



$\quad \delta_f^z \overline{(Wq^2l)}_b^z = \delta_b^z \left[ \frac{\overline{K_{q_f}}^z}{D^{t+1}} \delta_f^z (q^2l)^{t+1} \right] + lE_1 \frac{K_M}{D} \left\{ \left[ \delta_b^z (\overline{U}_f^x) \right]^2 + \left[ \delta_b^z (\overline{V}_f^y) \right]^2 \right\} \widetilde{W} +$
$\quad \frac{lE_1E_3g}{\rho_0} K_H \delta_b^z (\rho) \widetilde{W} - \frac{Dq^3}{B_1} + F_{q^2l}$ (B8)

Where $F_u$, $F_v$, $F_{q^2}$, and $F_{q^2l}$ are horizontal kinematic viscosity terms of u, v, $q^2$, and
$q^2l$, respectivly. $F_T$ and $F_S$ are horizontal diffusion terms of T and S respectivly.
$\quad F_u = \delta_b^x [2A_M D \delta_f^x (U^{t-1})] + \delta_f^y \left\{ \overline{(\overline{A_{M_b}}^x)_b}^y \overline{(\overline{D_b})_b}^y [\delta_b^x (V)^{t-1} + \delta_b^y (U)^{t-1}] \right\}$ (B9)
$\quad F_v = \delta_b^y [2A_M D \delta_f^y (V^{t-1})] + \delta_f^x \left\{ \overline{(\overline{A_{M_b}}^x)_b}^y \overline{(\overline{D_b})_b}^y [\delta_b^x (V)^{t-1} + \delta_b^y (U)^{t-1}] \right\}$ (B10)
$\quad F_T = \delta_f^x \left[ \overline{A_{H_b}}^x \overline{H_b}^x \delta_b^x (T^{t-1}) \right] + \delta_f^y \left[ \overline{A_{H_b}}^y \overline{H_b}^y \delta_b^y (T^{t-1}) \right]$ (B11)
$\quad F_S = \delta_f^x \left[ (\overline{A_{H_b}}^x \overline{H_b}^x \delta_b^x (S^{t-1}) \right] + \delta_f^y \left[ \overline{A_{H_b}}^y \overline{H_b}^y \delta_b^y (S^{t-1}) \right]$ (B12)
$\quad F_{q^2} = \delta_f^x \left[ \overline{(\overline{A_{M_b}}^x)_b}^z \overline{H_b}^x \delta_b^x (q^2)^{t-1} \right] + \delta_f^y \left[ \overline{\overline{A_{M_b}}^y_b}^z \overline{H_b}^y \delta_b^y (q^2)^{t-1} \right]$ (B13)
$\quad F_{q^2l} = \delta_f^x \left[ \overline{(\overline{A_{M_b}}^x)_b}^z \overline{H_b}^x \delta_b^x (q^2l)^{t-1} \right] + \delta_f^y \left[ \overline{\overline{A_{M_b}}^y_b}^z \overline{H_b}^y \delta_b^y (q^2l)^{t-1} \right]$ (B14)

The discrete governing equations of barotropic (external) mode expressed by operators
are shown as below:
$\quad \frac{\eta^{t+1} - \eta^{t-1}}{2dte} + \delta_f^x (\overline{D}_b^x \; U_A) + \delta_f^y (\overline{D}_b^y \; V_A) = 0$ (B15)
$\quad \frac{(\overline{D}_b^x U_A)^{t+1} - (\overline{D}_b^x U_A)^{t-1}}{2dte} + \delta_b^x \left[ \overline{(\overline{D}_b^x U_A)_f}^x \overline{(U_A)_f}^x \right] + \delta_f^y \left[ \overline{(\overline{D}_b^y V_A)_b}^x \overline{(U_A)_b}^y \right] -$
$\quad \overline{\left[ \tilde{f}_A \overline{(V_A)_f}^y D \right]}_b^x - \overline{\left[ f \overline{(V_A)_f}^y D \right]}_b^x + g \overline{D}_b^x \delta_b^x (\eta) = \delta_b^x \{ 2(AA_M) D \delta_f^x [(U_A)^{t-1}] \} +$
$\quad \delta_f^y \left\{ \overline{\left[ \overline{(AA_M)_b}^x \right]}_b^y \overline{(\overline{D}_b^x)_b}^y [\delta_b^x (V_A) + \delta_b^y (U_A)]^{t-1} \right\} + \phi_x$ (B16)
$\quad \frac{(\overline{D}_b^y V_A)^{t+1} - (\overline{D}_b^y V_A)^{t-1}}{2dte} + \delta_f^x \left[ \overline{(\overline{D}_b^x U_A)_b}^y \overline{(V_A)_b}^x \right] + \delta_b^y \left[ \overline{(\overline{D}_b^y V_A)_f}^y \overline{(V_A)_f}^y \right] +$





$\overline{\left[\tilde{f}_A\overline{(U_A)}_f^x D\right]}_b^y + \overline{\left[f\overline{(U_A)}_f^x D\right]}_b^y + g\overline{D}_b^y\delta_b^y(\eta) = \delta_b^y\{2(AA_M)D\delta_f^y[(V_A)^{t-1}]\} +$
$\delta_f^x\left\{\overline{\left[\overline{(AA_M)}_b^x\right]}_b^y \overline{(\overline{D}_b)}_b^x [\delta_b^x(V_A) + \delta_b^y(U_A)]^{t-1}\right\} + \phi_y$       (B17)

where
$\qquad\qquad \phi_x = -WU(0) + WU(-1) - \frac{g(\overline{D}_b^x)^2}{\rho_0}\int_{-1}^0\left\{\left[\int_\sigma^0 \delta_b^x\overline{(\rho)}_b^z d\sigma'\right]d\sigma\right\} +$
$\frac{g\overline{D}_b^x\delta_b^x D}{\rho_0}\int_{-1}^0\left\{\left[\int_\sigma^0 \overline{\sigma}_b^z\delta_b^z(\overline{\rho}_b^x)\right]d\sigma\right\} + G_x$       (B18)
$\qquad\qquad \phi_y = -WV(0) + WV(-1) - \frac{g(\overline{D}_b^y)^2}{\rho_0}\int_{-1}^0\left\{\left[\int_\sigma^0 \delta_b^y\overline{(\rho)}_b^z d\sigma'\right]d\sigma\right\} +$
$\frac{g\overline{D}_b^y\delta_b^y D}{\rho_0}\int_{-1}^0\left\{\left[\int_\sigma^0 \overline{\sigma}_b^z\delta_b^z(\overline{\rho}_b^y)\right]d\sigma\right\} + G_y$       (B19)


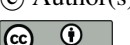


**Appendix C: Descriptions of symbols**


The description of each symbol in the governing equations is list as below:


Table C1. Descriptions of symbols

| Symbol | Description |
| --- | --- |
| $\eta$ | Free surface elevation |
| H | Bottom topography |
| ua, va | Vertical average velocity in x, y direction, respectively |
| U, V, W | Velocity in x, y, $\sigma$ direction, respectively |
| D | Fluid column depth |
| f | The Coriolis parameter |
| g | The gravitational acceleration |
| $\rho_0$ | Constant density |
| $\rho$ | Situ density |
| T | Potential temperature |
| S | Salinity |
| R | Surface solar radiation incident |
| $q^2/2$ | Turbulence kinetic energy |
| l | Turbulence length scale |
| $q^2l/2$ | Production of turbulence kinetic energy and turbulence length scale |
| dti | Time step of baroclinic mode |
| dte | Time step of barotropic mode |
| dx | Grid increment in x direction |
| dy | Grid increment in y direction |
| $A_M$ | Horizontal kinematic viscosity |
| $A_H$ | Horizontal heat diffusivity |
| $K_M$ | Vertical kinematic viscosity |
| $K_H$ | Vertical mixing coefficient of heat and salinity |
| $K_q$ | Vertical mixing coefficient of turbulence kinetic energy |




*Author contributions*. Xiaomeng Huang, Xing Huang, DW, QW, and SZ designed
OpenArray. Xing Huang, MW, YG, and QT implemented and tested GOMO.
Xiaomeng Huang and Xing Huang led the writing of this paper with contributions from
all other coauthors.

*Competing interests*. The authors declare that they have no conflict of interest.

*Acknowledgements*. Xiaomeng Huang is supported by a grant from the State's Key
Project of Research and Development Plan (2016YFB0201100) and the National
Natural Science Foundation of China (41776010). Xing Huang is supported by a grant
from the State's Key Project of Research and Development Plan (2018YFB0505000).
Shixun Zhang is supported by a grant from the State's Key Project of Research and
Development Plan (2017YFC1502200) and Qingdao National Laboratory for Marine
Science and Technology (QNLM2016ORP0108). Zhenya Song is supported by
National Natural Science Foundation of China (U1806205) and AoShan Talents
Cultivation Excellent Scholar Program Supported by Qingdao National Laboratory for
Marine Science and Technology (2017ASTCP-ES04).

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





**Tables**
Table 1. Definitions of the twelve basic operators

| Notations | Discrete Form | | | Basic Operator |
|---|---|---|---|---|
| $\overline{var}_f^x$ | [ $var(i,j,k)$ | + | $var(i+1,j,k)$ ] / 2 | AXF |
| $\overline{var}_b^x$ | [ $var(i,j,k)$ | + | $var(i-1,j,k)$ ] / 2 | AXB |
| $\overline{var}_f^y$ | [ $var(i,j,k)$ | + | $var(i,j+1,k)$ ] / 2 | AYF |
| $\overline{var}_b^y$ | [ $var(i,j,k)$ | + | $var(i,j-1,k)$ ] / 2 | AYB |
| $\overline{var}_f^z$ | [ $var(i,j,k)$ | + | $var(i,j,k+1)$ ] / 2 | AZF |
| $\overline{var}_b^z$ | [ $var(i,j,k)$ | + | $var(i,j,k-1)$ ] / 2 | AZB |
| $\delta_f^x(var)$ | [ $var(i+1,j,k)$ - | | $var(i,j,k)$ ] / $dx(i,j)$ | DXF |
| $\delta_b^x(var)$ | [ $var(i,j,k)$ | - | $var(i-1,j,k)$ ] / $dx(i-1,j)$ | DXB |
| $\delta_f^y(var)$ | [ $var(i,j+1,k)$ - | | $var(i,j,k)$ ] / $dy(i,j)$ | DYF |
| $\delta_b^y(var)$ | [ $var(i,j,k)$ | - | $var(i,j-1,k)$ ] / $dy(i,j-1)$ | DYB |
| $\delta_f^z(var)$ | [ $var(i,j,k+1)$ - | | $var(i,j,k)$ ] / $dz(k)$ | DZF |
| $\delta_b^z(var)$ | [ $var(i,j,k)$ | - | $var(i,j,k-1)$ ] / $dz(k-1)$ | DZB |







Table 2    The jumping rules of an operator acting on an *Array*

| The initial position of *var* | The position of [A/D]**X**[F/B] (var) | The position of [A/D]**Y**[F/B] (var) | The position of [A/D]**Z**[F/B] (var) |
|:---:|:---:|:---:|:---:|
| 0 | 1 | 2 | 4 |
| 1 | 0 | 3 | 5 |
| 2 | 3 | 0 | 6 |
| 3 | 2 | 1 | 7 |
| 4 | 5 | 6 | 0 |
| 5 | 4 | 7 | 1 |
| 6 | 7 | 4 | 2 |
| 7 | 6 | 5 | 3 |





Table 3. Comparing GOMO with several variations of the POM

| Model | Lines of code | Method | Computing Platforms |
|---|---|---|---|
| POM2k | 3521 | Serial | CPU |
| sbPOM | 4801 | MPI | CPU |
| mpiPOM | 9685 | MPI | CPU |
| POMgpu | 30443 | MPI + CUDA | GPU |
| GOMO | 1860 | OpenArray | CPU, Sunway |





Table. 4. Comparison of the amount of code for different functions

| Functions | Lines of code | | |
|---|---|---|---|
| | POM2k | sbPOM | GOMO |
| Solve for $\eta$ | 16 | 72 | **1** |
| Solve for Ua | 75 | 183 | **11** |
| Solve for Va | 75 | 183 | **11** |
| Solve for W | 36 | 90 | **3** |
| Solve for $q^2$ and $q^2l$ | 318 | 854 | **162** |
| Solve for T or S | 178 | 234 | **71** |
| Solve for U | 118 | 230 | **50** |
| Solve for V | 118 | 230 | **50** |





Table 5. Four benchmark tests

| Benchmark | Dimensions | Grid Size | OpenArray version (seconds) | Original version(seconds) |
|---|---|---|---|---|
| Continuity equation | 2D | 8192×8192 | 7.22 | 7.10 |
| Heat diffusion equation | 2D | 8192×8192 | 6.20 | 6.34 |
| Hotspot2D | 2D | 8192×8192 | 11.37 | 11.21 |
| Hotspot3D | 3D | 512×512×8 | 0.96 | 1.01 |







**Figures**

$1) 2D continuous equation

$$\eta_{t+1} = \eta_{t-1} - 2*dt*(\delta_f^x(\overline{D}_b^x*U) + \delta_f^y(\overline{D}_b^y*V))$$

$2) The code constructed by operators

elf=elb-2*dt*(DXF(AXB(D)*U)+DYF(AYB(D)*V))

$3) The pseudo-code

```
exchange2d_mpi(u,im,jm)
exchange2d_mpi(v,im,jm)
exchange2d_mpi(D,im,jm)

do i = 1, im
  do j = 1, jm
    elf(i,j)=elb(i,j)-2*dt*(        &
    ((D(i+1,j)+D(i,j))/2*u(i+1,j)-(D(i,j)+D(i-1,j))/2*u(i,j))/dx(i,j)+ &
    ((D(i,j+1)+D(i,j))/2*v(i+1,j)-(D(i,j)+D(i,j-1))/2*v(i,j))/dy(i,j))
```


Figure 1. Implementation of Eq. (4) by basic operators. The *elf* and *elb* are the surface
elevations at times (*t+1*) and (*t-1*) respectively.



*$ Equation (8)*

*elf=elb - 2\*dt\*( DXF( AXB(D)\*U ) + DYF( AYB(D)\*V ) )*

*$ Equation (9)*

*Uf=Db\*Ub/Df - 2\*dt/Df\*( DXB(AXF(AXB(D)\*U)\*AXF(U)) + DYF(AXB(AYB(D)\*V)\*AYB(U)) - &*
    *AXB(f\*AYF(V)\*D) + g\*AXB(D)\*DXB(el) - aam\*AXB(D)\*( DXB(DXF(Ub)) + DYF(DYB(Ub)) ) )*

*$ Equation (10)*

*Vf=Db\*Vb/Df - 2\*dt/Df\*( DXF(AYB(AXB(D)\*U)\*AXB(V)) + DYB(AYF(AYB(D)\*V)\*AYF(V)) + &*
    *AYB(f\*AXF(U)\*D) + g\*AYB(D)\*DYB(el) - aam\*AYB(D)\*( DXF(DXB(Vb)) + DYB(DYF(Vb)) ) )*


Figure 2. Implementation of the shallow water equations by basic operators. *elf*, *el* and
*elb* denote sea surface elevations at times $(t+1)$, $t$ and $(t-1)$, respectively. *Uf, U* and *Ub*
denote the zonal velocity at times $(t+1)$, $t$ and $(t-1)$, respectively. *Vf*, *V* and *Vb* denote
the meridional velocity at times $(t+1)$, $t$ and $(t-1)$, respectively. *aam* denotes the
viscosity coefficient.



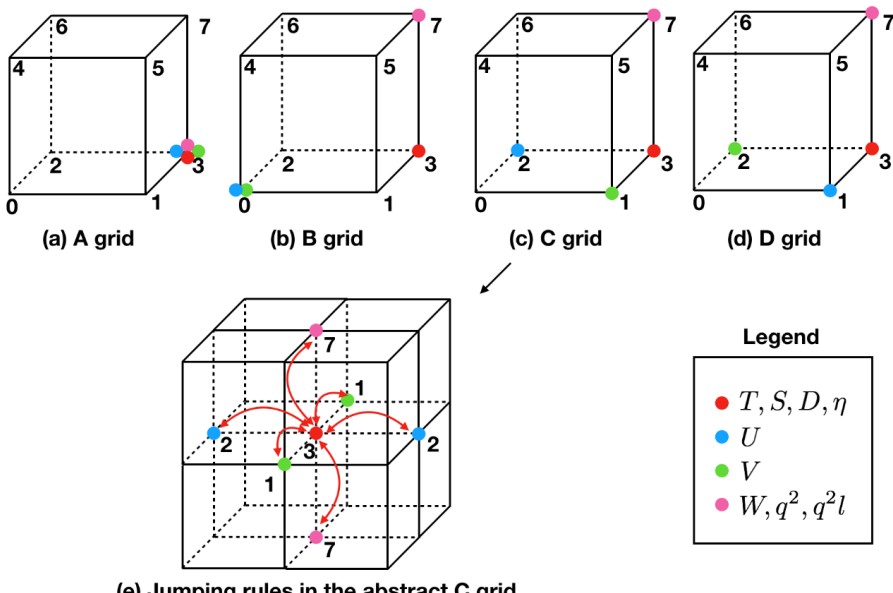

(e) Jumping rules in the abstract C grid


Figure 3. The schematic diagram of the relative positions of the variables on the

abstract staggered grid and the jumping procedures among the grid points.





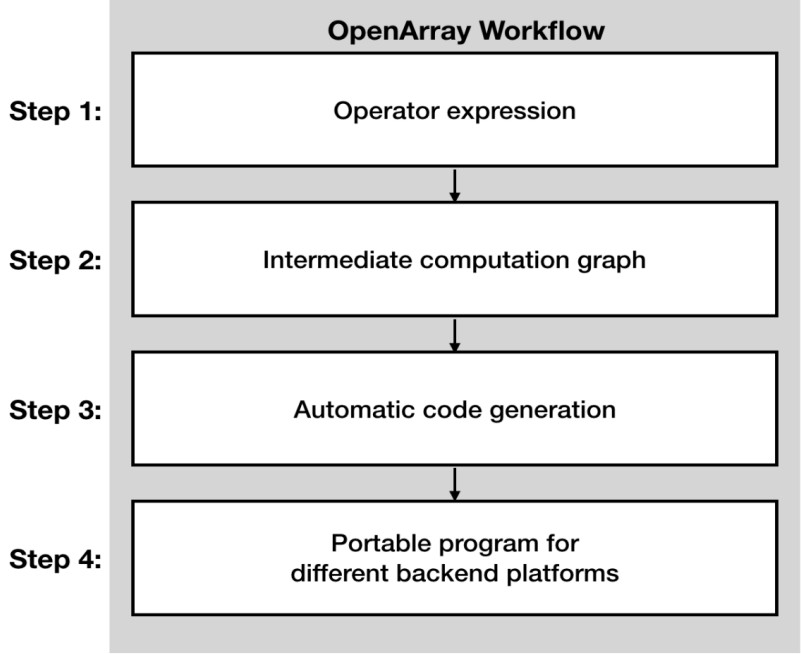


Figure 4. The workflow of OpenArray.




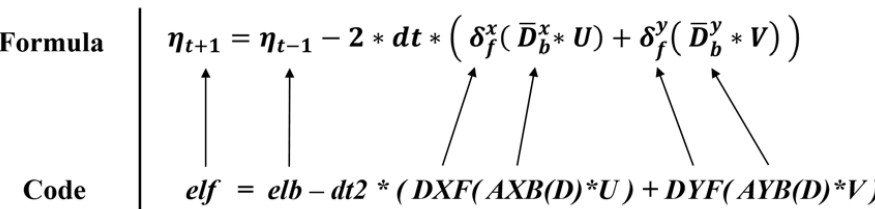

Figure 5. The effect of "The self-documenting code is the formula" illustrated by the

sea surface elevation equation.






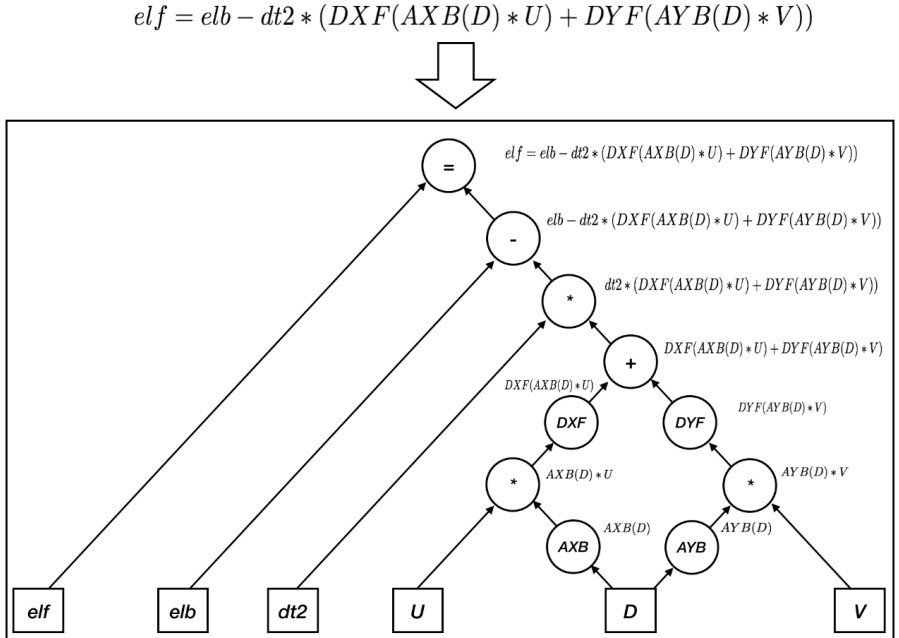


Figure 6. Parsing the operator expression form into the computation graph.



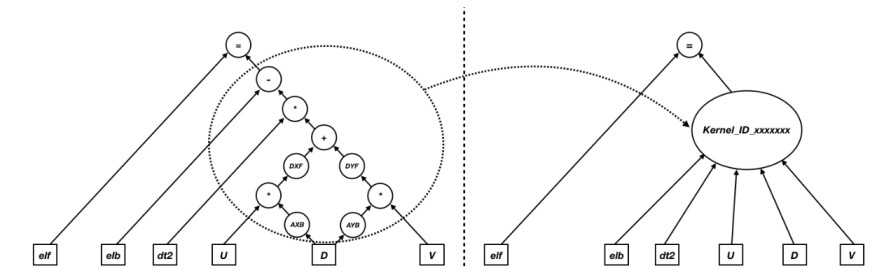


Figure 7. The schematic diagram of kernel fusion.






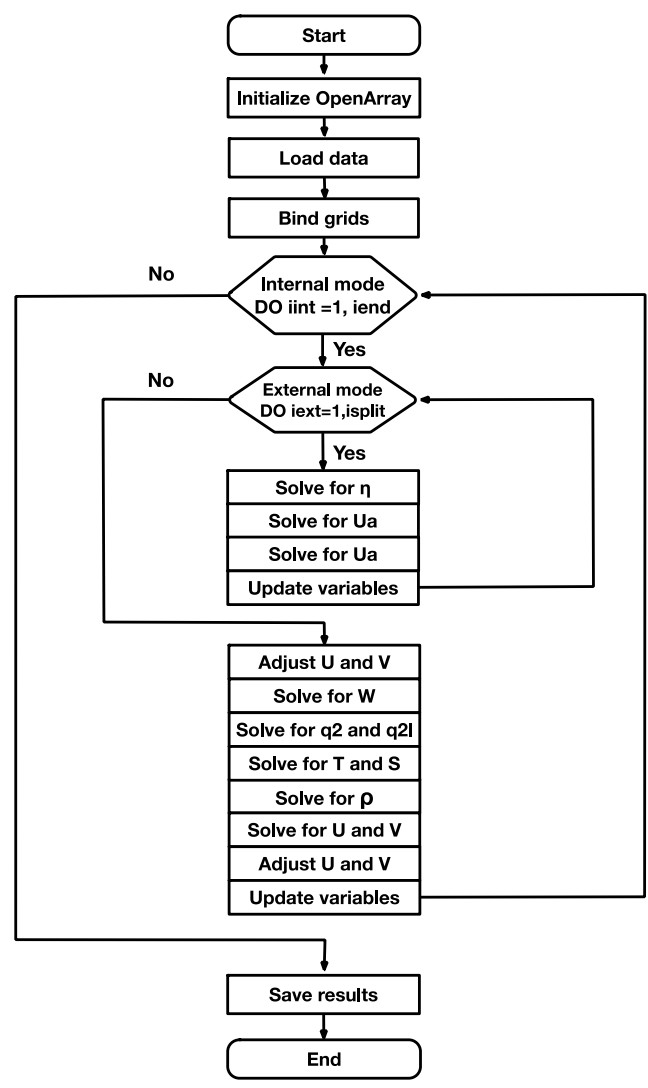


Figure 8. Flow diagram of GOMO






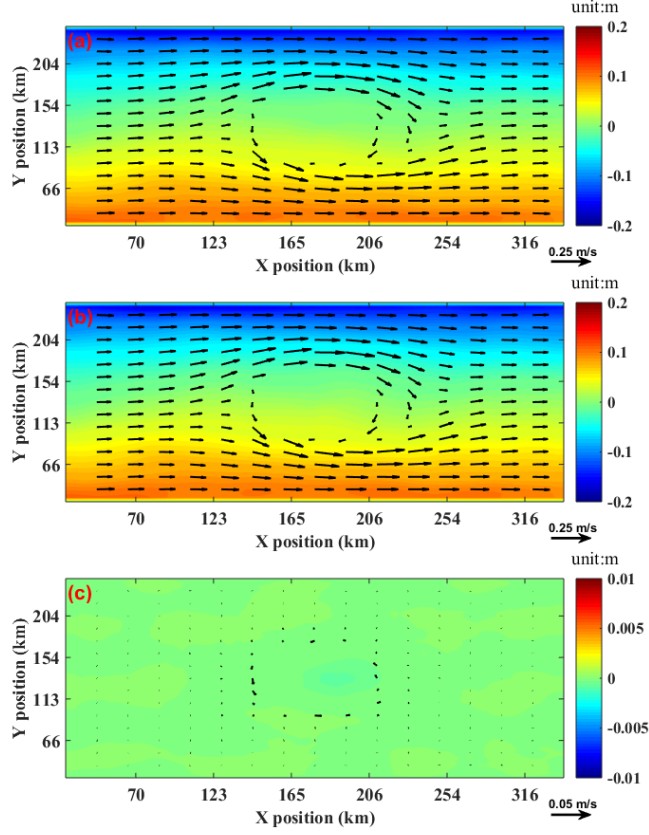


Figure 9. Comparison of the surface elevation (shaded) and currents at 3500 metres
depth (vector) between GOMO and sbPOM on the 4th model day. (a) GOMO, (b)
sbPOM, (c) GOMO-sbPOM.





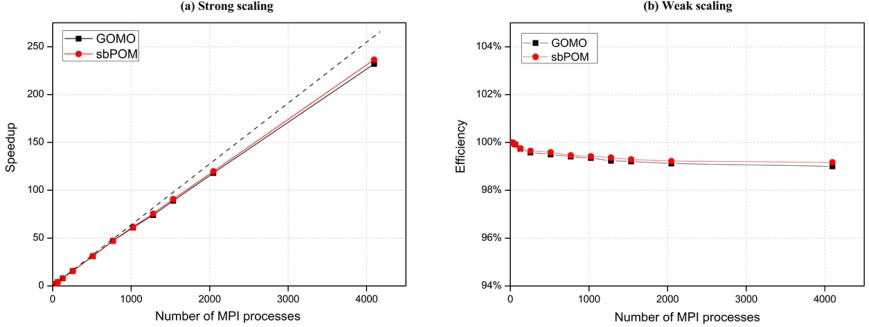


Figure 10. Performance comparison between sbPOM and GOMO. (a) The strong
scaling result; vertical axis denotes the speedup relative to 16 processes in a single node.
(b) The weak scaling result.





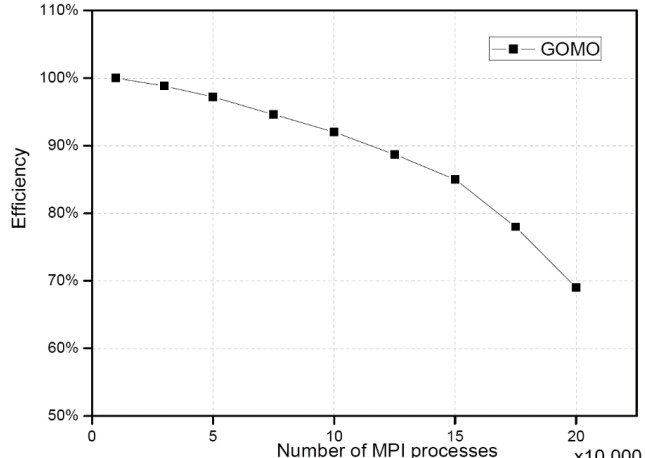


Figure 11. Parallel efficiency of GOMO on the Sunway TaihuLight supercomputer.