# Peer review of "1. Introduction"

_Geoscientific Model Development, 2019_

## Referee Comment (RC1) · Anonymous Referee #1 · 6 Mar 2019

OpenArray v1.0: A Simple Operator Library for the Decoupling of Ocean Modelling and Parallel Computing by Huang et al. describes a first implementation of a set of operators in a C++ library for writing a 3D ocean model. The basic concepts behind OpenArray are described and explained, and an initial set of operators is used to program a 3D ocean model to mimic the numerics of the Princeton Ocean Model (POM).

The technical aspects of this discussion paper are of broader interest to the climate modelling community, at least to those colleagues that are working more on the software engineering site rather than in pure geosciences. Despite my remarks and suggestions below, I find the OpenArray approach attractive, and the GOMO use case

makes the paper complete for a first release of the software and its publication.

I recommend a revised version of the paper for publication in Geoscientific Model Development.

P.S.: I like your conclusion, especially the last sentence.

**1    General Comments**

While the discussion paper is well structured and clearly written I am missing some parts which I outline below. In my opinion the publication as a whole does not need to be restructured or rewritten but I suggest to extent/rewrite/restructure the introduction by describing the state of the art in somewhat more detail. I wished the authors had mentioned their solution for IO and provided a discussion section to share their experiences and opinion about the pros and cons of their approach.

Unfortunately, I did not succeed to install OpenArray on an OSX system (macOS 10.14.3, Armadillo 9.2, Boost 1.66, LLVM 7.0, gcc/g++/gfortran 8.3, openmpi 3.0)

**1.1    Introduction**

While the general motivation to start the OpenArray approach is made clear in this paper I am missing a more complete discussion of the state of the art. A few approaches (ATMOL, ICON DSL, STELLA) are listed but the text does not provide useful hints in how far OpenArray really goes beyond existing approaches. I am missing ATLAS (DOI: 10.1016/j.jcp.2017.07.006) . I am not an expert in this field, but to me ATLAS seems to cover several design aspects, in particular operators, support for parallelism, and support for different grid types, and seems to be in these aspects similar to OpenArray. The ESCAPE project and its follow-up ESCAPE2 worked or will work in this direction. The

authors cite Lawrence et al., 2018 but only as a reference for a trend towards the usage of "heterogeneous and advanced computing platforms", even though Lawrence et al. also discuss software approaches to address these challenges, including concepts like those used by OpenArray.

**1.2   IO**

As a model without IO is pretty useless it would have been nice to have read a few lines about how the (parallel) IO is approached. It is included in OpenArray, so why not spending one paragraph on such an important issues as well, perhaps with some graph showing the IO performance.

**1.3   Discussion**

You are convincing in the sense that your approach is valid and takes major burden from the oceanographer who "only" wants to run and modify an ocean model. On the other hand the complexity has not magically disappeared but it is moved from GOMO (in this example) to OpenArray. When porting the whole software onto a new system the major effort now goes into OpenArray - which is fine, but it has to be done. How complex is this? How flexible is the OpenArray approach in this respect.

If the unlucky oceanographer comes up with the idea to try out yet another (perhaps higher-order) advection or any other scheme which is not yet supported by OpenArray, how difficult is it to extend OpenArray? Does this require expert knowledge and support from OpenArray developers?

How seamless is - in your opinion - the integration of other stencils into the OpenArray library?

I am not insisting on answering these questions line by line but rather take these as

suggestions for what could be addressed in a thorough discussion. You as authors may wish to stress different - and in your opinion more important - points.

**2 Specific Comments**

Line 39 and elsewhere: consider to replace "climate model" by "Earth system model", as the latter is now mainly used when talking about multi-component models in the context of Earth system and climate modelling efforts.

Line 41 ff: Please rephrase the sentence, as computing platforms are not applied but used.

Line 55: Which gap do you precisely have in mind? Do you really mean climate science in general or rather climate modelling (aka Earth system modelling)?

Line 70 ff: What is the former, what is the latter language? Could you briefly explain to the non-experts amongst the readers the difference between source-to-source and DSL? Perhaps this whole paragraph needs some restructuring (see remarks in my section 1.1).

Line 90: Please introduce the reader to the heterogeneity you have in mind here. What makes TaihuLight different in terms of heterogeneity? From the system specification further down in your text, TaihuLight does not look that heterogeneous.

Line 100: I would say that you solved the problem for one ocean model or a particular class of ocean models but not yet for ocean models in general.

Line 109: Is it really valid Fortran code, or shouldn't it better be classified as pseudo Fortran code as GOMO cannot really live without the OpenArray library?

Line 111: Is it really meant like that you compile and link one executable which can then be executed on any computing platform? Or are you talking about the intermediate C++

code? But this would require compiling and linking on the target system before it can be executed.

Line 120: True but only if the OpenArray has been ported to and is available on the Sunway platform. There is probably no free lunch when moving to a new hardware or software environment.

Line 148 and elsewhere: Equation 1 is probably taken from the POM user manual, but nevertheless should not expressions like $\frac{\partial DU}{\partial x}$ rather be written as $\frac{\partial}{\partial x}(DU)$ ?

Line 152: Could provide some hint on how to arrive at the discrete expression (2)

Line 211: I thought your current implementation of the operators only supports uniform (=equidistant) grids? What am I missing here?

Line 214 and elsewhere: I suggest to avoid the phrase "automatic". Nothing is done automatically but every effect has a cause. Here, something is happening because you programmed it that way and some conditions are coming together to trigger an action.

Line 238 ff: Could you clarify how the Arakawa grid type, the jumping rules and the differential operators are linked together? Let us assume I have formulated my ocean model on an Arakawa C grid and now for curiosity would like to run it on an A grid (not because it would really makes sense but to demonstrate the effect of discretisation on the numerical solution) what would I have to change in my ocean model code?

Line 247 ff: What is the motivation for the list of ocean model codes you provide in this paragraph? There are other codes around, e.g. FESOM (see https://fesom.de and the list of publication there) or an unstructured grid model for global ocean dynamics by P. Korn (see https://doi.org/10.1016/j.jcp.2017.03.009) and several more.

Line 274 section 3.1:

Could you add a few lines to describe the handling of lateral and vertical boundary conditions within your operators?

[Figure]

Line 334: When speaking of subgraphs and the kernel function, can the individual advection and diffusion terms be accessed for diagnostic purposes?

Line 352: I am not sure what is meant here and perhaps the sentence should be rephrased. Such a function needs to be programmed once for a particular ocean model, but once it is there it can be used, see e.g. ESMF_FieldBundleHalo contained in the Earth System Modeling Framework (ESMF). To my knowledge, other ocean models use similar approaches for the halo exchange as well. But no doubt, it is a relief to have it.

Line 391: Is the mode splitting algorithm inherited from POM, if so, this should be mentioned.

Line 422: Could you provide the number of lines for OpenArray as well? I could calculate it myself but . . .

Line 425: You raise the impression that porting is not an issue anymore. While this is certainly true for GOMO (which is of course very valuable) the porting still has to be done for OpenArray. Maybe this could be clarified somewhere (perhaps in the discussion).

Line 469 section 5.3: Which hardware do you use for these tests? What sets the upper bound of 4096 processes?

Line 481 section 5.4: What does this mean for a reasonable local domain size? 32X32 points as in sec. 5.3 is still on the good side, while 9x9 as used on TaihuLight does shows some performance degradation.

Line 490 ff: As I understand the steps up to compiling the JIT are done only once at the beginning of a job. If you run a longer experiment (in terms of wallclock time or number of timesteps) the initialisation phase should be negligible when compared to the total run time. Why don't you provide two numbers, one for the initialisation and one for the integration within the time loop?

**3 Technical Corrections/Suggestions**

L39 climate model → climate models

L42 climate community → climate modelling community

L43 model program needs → model programs need

L68 inefficiency → inefficiently

L83 the sentence needs to be reordered. It is not clear (to me) to which part "at the product level" is referring to.

L106 change to : . . . is similar to the original but manually optimized parallel program.

L108 support → supports

L125 → The implementation

L139 → In traditional ocean models . . .

L143 → When using the OpenArray library . . .

L211 → we propose

L220 → . . . the horizontal velocity components Array(U) and Array(V) are . . .

or

. . . the horizontal velocity Array(U, V) is . . .

L238 can be used → are used

L257 different → difference ; operator → operators

L289 will be concealed by → is hidden behind

L290 → . . . and can escape . . .
L303 → to implement a so-called lazy expression . . .

L318 . . . AYF are the interpolated functions → . . . AYF are the average functions.

L373 computing processing elements : aren't these Central Processing Units (CPUs)

L384 a practical ocean model → a numerical ocean model

L404 rephrase, the TKE and alike can be calculated but not the submodel.

L505 a practical ocean model → a numerical ocean model

---

## Author Comment (AC1) · 15 Mar 2019

**Dear editor and reviewer,**

**First of all, we would like to express our sincere appreciation to your valuable feedbacks. Your comments are highly insightful and enable us to substantially improve the quality of our manuscript and OpenArray. Below are our point-by-point responses to all the comments and our plans to revise the manuscript.**

**Responses to the comments of referee #1**

**1 General Comments**

While the discussion paper is well structured and clearly written I am missing some parts which I outline below. In my opinion the publication as a whole does not need to be restructured or rewritten but I suggest to extent/rewrite/restructure the introduction by describing the state of the art in somewhat more detail. I wished the authors had mentioned their solution for IO and provided a discussion section to share their experiences and opinion about the pros and cons of their approach.
**[Response]:**
We appreciate for your helpful suggestions. First, we will restructure the introduction and describe the state of art in more details, especially by adding comparisons to similar work. Second, we will provide more details of the I/O frameworks, including implementation method and future improvement plan. Lastly, we will add a discussion section to present our experiences and opinion about the pros and cons of our approach.

Unfortunately, I did not succeed to install OpenArray on an OSX system (macOS 10.14.3, Armadillo 9.2, Boost 1.66, LLVM 7.0, gcc/g++/gfortran 8.3, openmpi 3.0).
**[Response]:**
Sorry to hear that. First, we would like to repeat the following requirements.
Before compiling OpenArray, the following dependent libraries are required:
- Fortran 90 or Fortran 95 compiler.
- gcc/g++ compiler version 6.1.0 or higher.
- Intel icc/icpc compiler version 2017 or higher.
- GNU make version 3.81 or higher.
- Parallel NetCDF library version 1.7.0 or higher.
- Message Passing Interface (MPI) library.
- Armadillo, a C++ library for linear algebra & scientific computing, version 8.200.2 or higher.
- Boost C++ Libraries, version 1.65.1 or higher.
- LLVM compiler version 6.0.0 or higher.

After the required libraries are installed successfully, checkout to branch dev and type "./test.sh" in the home directory of OpenArray, the source code will be generated in the

build folder. Then, change the directory into build, type "make -f makefile.intel oalib_obj", then libopenarray.a and openarray.mod will be generated if there is no other problem.

According to the information you provided, a Parallel NetCDF library and a Message Passing Interface (MPI) library are needed. If you have any question, please contact us at any time.

**1.1 Introduction**

While the general motivation to start the OpenArray approach is made clear in this paper I am missing a more complete discussion of the state of the art. A few approaches (ATMOL, ICON DSL, STELLA) are listed but the text does not provide useful hints in how far OpenArray really goes beyond existing approaches. I am missing ATLAS (DOI: 10.1016/j.jcp.2017.07.006) . I am not an expert in this field, but to me ATLAS seems to cover several design aspects, in particular operators, support for parallelism, and support for different grid types, and seems to be in these aspects similar to OpenArray. The ESCAPE project and its follow-up ESCAPE2 worked or will work in this direction. The authors cite Lawrence et al., 2018 but only as a reference for a trend towards the usage of "heterogeneous and advanced computing platforms", even though Lawrence et al. also discuss software approaches to address these challenges, including concepts like those used by OpenArray.

**[Response]:**
Thanks for your nice suggestions. We will rewrite the introduction part, introducing more details about the related work, including ATMOL, ICON DSL, STELLA and ATLAS. In addition, we will compare OpenArray with them in detail to demonstrate the advantages and disadvantages of OpenArray in the revised manuscript.

**1.2 IO**

As a model without IO is pretty useless it would have been nice to have read a few lines about how the (parallel) IO is approached. It is included in OpenArray, so why not spending one paragraph on such an important issues as well, perhaps with some graph showing the IO performance.

**[Response]:**
Thanks for your valuable suggestions. We will add an introduction to the existing IO frameworks, implementation methods and performance. Actually, at present we encapsulated the PnetCDF library to perform IO functions. However, in previous work, we developed CFIO, a fast I/O library for climate models published in Geoscientific Model Development as well (X. Huang, doi:10.5194/gmd-7-93-2014). In the future, we plan to merge CFIO into OpenArray to obtain better IO performance.

**1.3 Discussion**

You are convincing in the sense that your approach is valid and takes major burden from the oceanographer who "only" wants to run and modify an ocean model. On the other hand the complexity has not magically disappeared but it is moved from GOMO (in this example) to OpenArray. When porting the whole software onto a new system the major effort now goes into OpenArray - which is fine, but it has to be done. How complex is this? How flexible is the OpenArray approach in this respect.

**[Response]:**

Thanks for your helpful comments. As you pointed out, we moved the complexity from GOMO to OpenArray. Thus, the major burden of code porting is taken away for the oceanographers. OpenArray is designed to support multiple hardware platforms through separating hardware-dependent functions such as low-level numerical computations from the main framework. In section 3.4, we will provide more details to describe the complexity of porting OpenArray to Sunway platform, including methods, workload, difficulties, etc. In the future, OpenArray will support more categories of hardware platforms.

If the unlucky oceanographer comes up with the idea to try out yet another (perhaps higher-order) advection or any other scheme which is not yet supported by OpenArray, how difficult is it to extend OpenArray? Does this require expert knowledge and support from OpenArray developers?

**[Response]:**

Thanks for your comments. In the current version of OpenArray, we implemented the functions of 12 basic operators and the rules for generating fusion-kernel. If users want to extend OpenArray, support from OpenArray developers is required at present. In our future work, OpenArray will be extended to support the customized operators. The definition of operators will be written in the configuration file, and the pre-compiled template will be used to generate the operator function. After that, adding the customized operators no longer require expert knowledge and support from OpenArray developers.

How seamless is - in your opinion - the integration of other stencils into the OpenArray library?

**[Response]:**

We use a template mechanism to implement the stencil operators, in which the rules of stencil operations are defined in a separate file, so it is easy to generate any kind of stencil kernels. The difficult part is the communication pattern among processes, for example C[i] = A[i] + B[i-2], an efficient mechanism to manage the data transfer for customize stencils are yet to be developed. The integration of other stencils into the current version of OpenArray library still requires the support of OpenArray developers. The customized operators are under development. In the future, the next version of OpenArray will support customized operators, other stencils will be conveniently

integrated into OpenArray even without the support of OpenArray developers.

I am not insisting on answering these questions line by line but rather take these as suggestions for what could be addressed in a thorough discussion. You as authors may wish to stress different - and in your opinion more important - points.

**2 Specific Comments**

Line 39 and elsewhere: consider to replace "climate model" by "Earth system model", as the latter is now mainly used when talking about multi-component models in the context of Earth system and climate modelling efforts.
**[Response]:**
Thanks for the suggestion. In the revised manuscript, we have replaced "climate model" by "Earth system model" in Line 18, Line 31, Line 39, Line 52, Line 57, and Line 83.

Line 41 ff: Please rephrase the sentence, as computing platforms are not applied but used.
**[Response]:**
Thanks for the suggestion. We have replaced "applied" with "used" in the sentence at Line 42.

Line 55: Which gap do you precisely have in mind? Do you really mean climate science in general or rather climate modelling (aka Earth system modelling)?
**[Response]:**
Sorry for the confusion. The idea of gap or chasm is quoted from the reference (Bryan N. Lawrence et al, 2018, https://doi.org/10.5194/gmd-11-1799-2018). Due to the ever-increasing complexity of earth system models coupled with rapidly evolving computational techniques, climate modelling will arrive at a gap which will separate scientific aspiration from our ability to develop and/or rapidly adapt codes to the available hardware. We have replaced "climate science" with "climate modelling" at Line 42.

Line 70 ff: What is the former, what is the latter language? Could you briefly explain to the non-experts amongst the readers the difference between source-to-source and DSL? Perhaps this whole paragraph needs some restructuring (see remarks in my section 1.1).
**[Response]:**
Thanks for your comments. We will restructure this whole paragraph and add following sentences to explain the differences between source-to-source and DSL.

"Source-to-source translator can parse a piece of code as input and generate a novel code as output, such as from Fortran to C, or take a high-level language as input and convert the code with parallel code comments. While DSLs are developed to meet the growing needs of the development of applications. For example, Partitioned Global

Address Space (PGAS) takes the local memory in different processes as a global memory space and make network communication invisible to the programmers. Many advanced programing languages emerged from this concept, such as Coarray Fortran (Numrich et al, 1998; Mellor-Crummey et al., 2009), Unified Parallel C (El-Ghazawi and Smith, 2006), X10 (Charles et al., 2005), Chapel (Chamberlain et al., 2007) and XcalableMP (Nakao et al., 2012). Some other outstanding DSLs, such as OpenFoam (Jasak et al, 2007), ATMOL (Engelen, 2002), ICON DSL (Torres et al, 2013), and STELLA (Gysi et al, 2015), provide high-level abstraction interface close to the mathematical notations for domain scientists to write much more concise and simpler code."

Line 90: Please introduce the reader to the heterogeneity you have in mind here. What makes TaihuLight different in terms of heterogeneity? From the system specification further down in your text, TaihuLight does not look that heterogeneous.
**[Response]:**
Sorry for the confusion. One major technology innovation of the Sunway TaihuLight supercomputer is the homegrown SW26010 heterogeneous many-core processor. It includes four powerful core groups (CGs) connected via the network on chip (NoC), each of CGs consists of a manage processing element (MPE) and 64 computing processing elements (CPEs) arranged in an eight by eight mesh. We will add more details about the architecture of Sunway TaihuLight in the revised manuscript.

Line 100: I would say that you solved the problem for one ocean model or a particular class of ocean models but not yet for ocean models in general.
**[Response]:**
Thanks for your corrections. We have added "a particular class of" in front of "ocean models using the finite difference method and staggered grid in OpenArray." (Line 100)

Line 109: Is it really valid Fortran code, or shouldn't it better be classified as pseudo Fortran code as GOMO cannot really live without the OpenArray library?
**[Response]:**
Sorry for the confusion. OpenArray works as an independent library. As you pointed out, OpenArray is the essential for GOMO, since GOMO uses the functions and modules provided by OpenArray, such as average operators, differential operators, assignment functions, I/O functions, et al. While, GOMO is written by valid Fortran codes.

Line 111: Is it really meant like that you compile and link one executable which can then be executed on any computing platform? Or are you talking about the intermediate C++ code? But this would require compiling and linking on the target system before it can be executed.
**[Response]:**
Sorry for the confusion. As you mentioned, there is no free lunch in the world. We transfer the cross-platform complexity to OpenArray. When porting the ocean models

to a new platform, we need to redesign an additional function for the target system and add it to the code generation module in OpenArray. This added function is used to translate the intermediate computation graph into corresponding executable code for the target platform. OpenArray supports X86 and Sunway platforms, therefore GOMO is executable on the two platforms without additional modification.

We have changed the sentence "The final output is a program that is executable on different computing platforms." into "The final output is a program that is executable on the two computing platforms currently supported by OpenArray". (Line 111~112)

Line 120: True but only if the OpenArray has been ported to and is available on the Sunway platform. There is probably no free lunch when moving to a new hardware or software environment.
**[Response]:**
Thanks for your comments. As you mentioned, there is no free lunch when moving to a new hardware or software environment. In fact, we transfer the burden of porting code from GOMO to OpenArray, thus decoupling the ocean models from the hardware platforms is possible.

Line 148 and elsewhere: Equation 1 is probably taken from the POM user manual, but nevertheless should not expressions like $\frac{\partial DU}{\partial x}$ rather be written as $\frac{\partial}{\partial x}(DU)$ ?

**[Response]:**
Thanks for your suggestions. As you pointed out, the Eq. 1 is indeed derived from the POM user manual. Therefore, we would like to use the same expressions of the equations with that in POM user manual (shown as the following equation).

$$\frac{\partial DU}{\partial x} + \frac{\partial DV}{\partial y} + \frac{\partial \omega}{\partial \sigma} + \frac{\partial \eta}{\partial t} = 0$$

Line 152: Could provide some hint on how to arrive at the discrete expression (2).
**[Response]:**
Thanks for your suggestions. In this paragraph, we will provide the details of Arakawa C grid and finite difference method to demonstrate the process from Eq. (1) to the discrete Eq. (2).

In Arakawa C grid, fluid depth $D$ is calculated at the centers. $U$ component is calculated at the left and right side of the variable $D$, $V$ component is calculated at the lower and upper side of the variable $D$ (Shown as Fig. 1). Taking the term $\frac{\partial DU}{\partial x}$ for an example, we firstly apply linear interpolation to obtain the $D's$ value at $U$ point represented by $tmp1$. Through a backward difference to the product of $tmp1$ and $U$, then the discrete expression of $\frac{\partial DU}{\partial x}$ can be obtained.

$tmp1 = 0.5*(D(i+1,j)+D(i,j))*U(i+1,j)$

$$\frac{\partial DU}{\partial x} = \frac{0.5 * \big(D(i+1,j)+D(i,j)\big) * U(i+1,j) - 0.5 * \big(D(i,j)+D(i-1,j)\big) * U(i,j)}{dx(i,j)^*}$$

Where $dx(i,j)^*=0.5*( dx(i,j) + dx(i-1,j) )$

Unfortunately, we missed 0.5 in the Eq.2 (Line 153~154). We have changed the Eq.2 into the following form.

$$\frac{\eta_{t+1}(i,j) - \eta_{t-1}(i,j)}{2 * dt} + \frac{0.5 * \big(D(i+1,j)+D(i,j)\big) * U(i+1,j) - 0.5 * \big(D(i,j)+D(i-1,j)\big) * U(i,j)}{dx(i,j)^*}$$

$$+ \quad \frac{0.5 * \big(D(i,j+1)+D(i,j)\big) * V(i,j+1) - 0.5 * \big(D(i,j)+D(i,j-1)\big) * V(i,j)}{dy(i,j)^*} = 0$$

Where $dx(i,j)^*=0.5*( dx(i,j) + dx(i-1,j) )$, $dy(i,j)^*=0.5*( dy(i,j) + dy(i,j-1) )$

[Figure]

Figure 1. Arrangement of variables in the staggered Arakawa C grid.

Line 211: I thought your current implementation of the operators only supports uniform (=equidistant) grids? What am I missing here?
**[Response]:**
Sorry for the confusion. In OpenArray, grid increments (*dx, dy, dz*) are combined with physical variables through grid binding. On the staggering Arakawa C-grid, the components of velocity (*u, v, w*) and potential temperature (*T*) et al are defined at different points. Every variable on each point has its own set of grid increments so the current implementation of the operators supports varying grids.

Line 214 and elsewhere: I suggest to avoid the phrase "automatic". Nothing is done automatically but every effect has a cause. Here, something is happening because you programmed it that way and some conditions are coming together to trigger an action.
**[Response]:**
Thanks for your corrections. We have replaced all the phrases "automatic/automatically" with "implicit/implicitly" in the revised manuscript. (Line 22, 29, 72, 74, 110, 214, 238,

268, 355)

Line 238 ff: Could you clarify how the Arakawa grid type, the jumping rules and the differential operators are linked together? Let us assume I have formulated my ocean model on an Arakawa C grid and now for curiosity would like to run it on an A grid (not because it would really makes sense but to demonstrate the effect of discretization on the numerical solution) what would I have to change in my ocean model code?

**[Response]:**

Thanks for your valuable suggestion. To make it clearer, we will take Eq. 1 as an example to demonstrate the effect of the numerical solution based on Arakawa A grid, B grid, and C grid and illustrate the modifications to the ocean models when grid scheme is changed in the revised manuscript.

Line 247 ff: What is the motivation for the list of ocean model codes you provide in this paragraph? There are other codes around, e.g. FESOM (see https://fesom.de and the list of publication there) or an unstructured grid model for global ocean dynamics by P. Korn (see https://doi.org/10.1016/j.jcp.2017.03.009) and several more.

**[Response]:**

Sorry for the confusion. In this paragraph, we want to express our respect to these excellent ocean models, such as POM, ROMS, MITgcm, FESOM et al. Considering most of these existing ocean models use finite difference or finite volume methods on structured meshes, we designed 12 basic operators in OpenArray only for this particular class of ocean models at present. In our future work, more customized operators will be implemented to support other numerical methods and meshes.

We have changed "…, including Advanced Circulation model (ADCIRC) …" into "…, including Finite-Element/volumE Sea ice-Ocean Mode (FESOM) (Korn, 2017), Advanced Circulation model (ADCIRC) …"

Line 274 section 3.1:

Could you add a few lines to describe the handling of lateral and vertical boundary conditions within your operators?

**[Response]:**

Thanks for your comments. In section 3.1, we will add the details to describe how to handle the lateral and vertical boundary conditions within the operators. First, the lateral and vertical boundary conditions are set to zeros within the operators and operator expressions. Then, GOMO invokes the modules which is independent with operators to update the boundary conditions.

Line 334: When speaking of subgraphs and the kernel function, can the individual advection and diffusion terms be accessed for diagnostic purposes?

**[Response]:**

Thanks for your comments. If users want access to individual variables or subgraphs, they need to split the formula or code into multiple expressions for diagnostic or

printing purposes.

Line 352: I am not sure what is meant here and perhaps the sentence should be rephrased. Such a function needs to be programmed once for a particular ocean model, but once it is there it can be used, see e.g. ESMF_FieldBundleHalo contained in the Earth System Modeling Framework (ESMF). To my knowledge, other ocean models use similar approaches for the halo exchange as well. But no doubt, it is a relief to have it.
**[Response]:**
Thanks for your suggestions, the sentence will be rephrased (Line 352). As you pointed out, earth system models generally call the particular functions to control the communication between the local boundary regions, such ESMF_FieldBundleHalo in ESMF. In OpenArray, we further hide these procedures of communication through the fused kernel. Users do not need to explicitly call these functions (e.g. ESMF_FieldBundleHalo) for the halo exchange, or know the parallel details.

Line 391: Is the mode splitting algorithm inherited from POM, if so, this should be mentioned.
**[Response]:**
Thanks for your suggestion. As you pointed out, the mode splitting algorithm is inherited from POM. We have changed "… the mode-splitting algorithm to address …" into "… the mode-splitting algorithm inherited from POM to address …" (Line 391)

Line 422: Could you provide the number of lines for OpenArray as well? I could calculate it myself but . . .
**[Response]:**
Thanks for your suggestion. The code of OpenArray is about 11,800 lines. We will add the number of lines for OpenArray into the sentence. (Line 422)

Line 425: You raise the impression that porting is not an issue anymore. While this is certainly true for GOMO (which is of course very valuable) the porting still has to be done for OpenArray. Maybe this could be clarified somewhere (perhaps in the discussion).
**[Response]:**
Thanks for your comments. As you pointed out, there is no free lunch when moving to a new hardware or software environment. Since the porting has been transferred to OpenArray, the models can be run on these platforms supported by OpenArray. We will clarify this in the discussion section in our revised manuscript.

Line 469 section 5.3: Which hardware do you use for these tests? What sets the upper bound of 4096 processes?
**[Response]:**
Thanks for your comments. We used the x86 cluster at National Supercomputing Center in Wuxi of China, which provides 5000 Intel Xeon E5-2650 v2 CPUs at most. In the future, we will increase the upper bound if more computing resources are available.

Line 481 section 5.4: What does this mean for a reasonable local domain size? 32X32 points as in sec. 5.3 is still on the good side, while 9x9 as used on TaihuLight does shows some performance degradation.

**[Response]:**

Thanks for your comments. On the same machine architecture, the parallel efficiency will be better if the local domain size is bigger. The architecture of Sunway TaihuLight is different from x86 machine architecture, so we did not compare and analyze the parallel scalability between them.

Line 490 ff: As I understand the steps up to compiling the JIT are done only once at the beginning of a job. If you run a longer experiment (in terms of wallclock time or number of timesteps) the initialisation phase should be negligible when compared to the total run time. Why don't you provide two numbers, one for the initialisation and one for the integration within the time loop?

**[Response]:**

Thanks for your suggestion. As you mentioned, the fusion-kernel codes are generated and compiled only once at the beginning of a job. In GOMO, this initialization phase consumes about 1 minute. In our revised manuscript, we will provide the consuming time for the initialization and the integration within the time loop, respectively. (Line 490)

**3 Technical Corrections/Suggestions**

L39 climate model → climate models
**[Response]:**
Corrected.

L42 climate community → climate modelling community
**[Response]:**
Corrected.

L43 model program needs → model programs need
**[Response]:**
Corrected.

L68 inefficiency → inefficiently
**[Response]:**
Corrected.

L83 the sentence needs to be reordered. It is not clear (to me) to which part "at the product level" is referring to.
**[Response]:**

Sorry for the confusion. To make a clear illustration, we have removed "at the product level", and changed the sentence "…, one major difficulty is the requirement of a stable and robust compiler, rather than an experimental compiler, at the product level" into "…, one major difficulty is the requirement of a stable, reliable, and robust compiler with good support". (Line 83~84)

L106 change to : . . . is similar to the original but manually optimized parallel program.
**[Response]:**
Corrected.

L108 support → supports
**[Response]:**
Corrected.

L125 → The implementation
**[Response]:**
Corrected.

L139 → In traditional ocean models . . .
**[Response]:**
Corrected.

L143 → When using the OpenArray library . . .
**[Response]:**
Corrected.

L211 → we propose
**[Response]:**
Corrected.

L220 → . . . the horizontal velocity components Array(U) and Array(V) are . . .    or
            . . . the horizontal velocity Array(U, V) is . . .
**[Response]:**
Thanks for your corrections. We have changed the sentence ". . . the horizontal velocity Array(U, V) are . . ." into ". . . the horizontal velocity Array(U, V) is . . ." (Line 220)

L238 can be used → are used
**[Response]:**
Corrected.

L257 different → difference ; operator → operators
**[Response]:**
Corrected.

L289 will be concealed by → is hidden behind
**[Response]:**
Corrected.

L290 → : : : and can escape . . .
**[Response]:**
Corrected.

L303 → to implement a so-called lazy expression . . .
**[Response]:**
Corrected.

L318 . . .AYF are the interpolated functions → . . .AYF are the average functions.
**[Response]:**
Corrected.

L373 computing processing elements: aren't these Central Processing Units (CPUs)
**[Response]:**
Sorry for the confusion.

In contrast with other existing heterogeneous supercomputers, which include both CPU processors and PCIe-connected many-core accelerators (NVIDIA GPU or Intel Xeon Phi), the computing power of TaihuLight is provided by a homegrown many-core SW26010 processor (or SW26010 CPU). The processor includes four core-groups (CGs). Each CG includes one management processing element (MPE), one computing processing element (CPE) cluster with eight by eight CPEs, and one memory controller (MC) (Shown as Fig. 2).

The MPE is a complete 64-bit RISC core, which can run in both the user and system modes. While, the CPE is also a 64-bit RISC core, but with limited functions. The CPE can only run in user mode and does not support interrupt functions. The CPE is designed to achieve the maximum aggregated computing power, while minimizing the complexity of the micro-architecture.

[Figure]

Figure 2. The architecture of the Sunway processor
Note: The paragraphs above and Fig. 2 are quoted from the reference (Haohuan Fu et al, 2016, https://link.springer.com/article/10.1007/s11432-016-5588-7).

To make a clear illustration, we will add more detailed introduction to the Sunway TaihuLight supercomputer in the revised manuscript.

L384 a practical ocean model → a numerical ocean model
**[Response]:**
Corrected.

L404 rephrase, the TKE and alike can be calculated but not the submodel.
**[Response]:**
Thanks for the suggestion. We have changed the sentence "…, and turbulence closure sub-model (q2, q2l) (Mellor and Yamada, 1982)" into "…, and turbulence closure scheme (q2, q2l) (Mellor and Yamada, 1982)".

L505 a practical ocean model → a numerical ocean model
**[Response]:**
Corrected.

We really appreciate your highly constructive comments.

Best wishes,
Xiaomeng Huang, Xing Huang, Dong Wang, Qi Wu et al.

---

## Referee Comment (RC2) · David Webb (Referee) · 25 Mar 2019

This is a well written paper concerned with generating an ocean model from a set of equations closer in form to the underlying differential equations than is usual. It is an interesting computational exercise but the resulting code has some important deficiencies and for that reason I think it would be more suited for a computational journal than the present one.

My main concerns all involve computational efficiency. As the authors state, ocean models are memory bandwidth limited and for that reason the code is usually written in such a way that once a variable is in one of the processor registers or in the highest

speed cache it is used as much as possible before being replaced. In fact the aim should be never to move variables more than once each timestep. In the present code the authors spend a major section reporting on one small step in this direction, but really this should only be the first of many steps.

I am also concerned about the way the code deals with multi-processor architectures. Unfortunately although I was able to compile both the c++ and fortran sections, the link step failed and so I was not able to check the running code. However the main time stepping loop appears to run on a single processor and it is only the difference and averaging operators, in the c++ code, which make use of the multi-processor architecture. This is surprising given the authors emphasis on the importance of parallel computing.

I am also a bit wary about all the details being lost in the c++ compiler/interpreter code. The authors emphasis the possibility of portability but this implies a large organisation continually keeping such a code up to date and adapted to the latest hardware. If not, the climate modelling groups have to do it themselves in which case the effort required will be much the same as now except for the addition of the compiler/interpreter and associated packages.

Both of the architectures discussed appear to be cache based but as I understand it the next major advance will come from better use of gpus. In these systems I expect variables will stay in gpu memory or be swapped between gpus and rarely return to the main cpu memory. For such systems the structure proposed here appears unsuitable.

There is also the question of how researchers might try out new code with the prosed library and debug the resulting runs. No user manual is provided and it is difficult to see how bugs can be traced, especially if they involve the c++ section of the code.

I also do not understand why a just-in-time compiler is used, given that the model grids do not change in time so that both human and computer effort would be better spent optimising the fixed grid code. And on this theme I am also concerned, although I would

like to be proved wrong, that what has been achieved here is little different from what might be achieved with a good fortran coarray code, together with statement functions or cpp define statements to take the place of the operators.

One reason that these are not used in ocean models comes from the fact that in a typical ocean model only about half the theoretical 3-D grid is involved in the calculation, the rest representing land or ocean topography. When computer memory and power is readily available a factor of two does not really matter but given the computational cost of many ocean models, spending time on such cells, as usually happens when coarays are used, can be critical.

Anyway - you can see that I am unhappy. However I must emphasise that I can also see that the paper represents a lot of hard work and I accept that as an example of an attempt to tackle good computational problem it is worthwhile. For this reason I think that publication in a journal more closely linked to the development of artificial intelligence would be more suitable.

If on the basis of the other referees reports, the authors are asked to provide a revised version then there are two extra documents I would like to see in the additional documentation section. The first is a user manual. The second is a full list of the include files and libraries needed (i.e. all those which are not the fortran or c++ compiler files) - to help with the problems I had.

In such a case I would also like to see some good answers to the concerns detailed above and, although I have not mentioned it so far, a few fewer buzz-phrases.

David Webb.

―――――――――――――――――

---

## Author Comment (AC2) · 16 Apr 2019

Dear Dr. Webb,

First of all, we would like to express our sincere appreciation to your valuable feedbacks. Your comments help us to substantially improve the quality of our manuscript and OpenArray. We uploaded two pdf files, one includes our point-by-point responses to your comments and our plans to revise the manuscript. The other is a simple user mannual of OpenArray. For more details, please refer to the supplement.

Please also note the supplement to this comment:

[Figure]

https://www.geosci-model-dev-discuss.net/gmd-2019-28/gmd-2019-28-AC2-supplement.zip

---

## Author Response (AR1)

Dear editor and reviewers,

First of all, we would like to express our sincere appreciation to your valuable feedbacks. Your comments are highly insightful and enable us to substantially improve the quality of our manuscript and OpenArray. Below are our point-by-point responses to all comments.

**1 Responses to the comments of referee #1**

**1.1 General Comments**

While the discussion paper is well structured and clearly written I am missing some parts which I outline below. In my opinion the publication as a whole does not need to be restructured or rewritten but I suggest to extent/rewrite/restructure the introduction by describing the state of the art in somewhat more detail. I wished the authors had mentioned their solution for IO and provided a discussion section to share their experiences and opinion about the pros and cons of their approach.

**[Response]:**
We appreciate for your helpful suggestions. First, we have restructured the introduction section and described the state of art in more details, especially by adding comparisons to similar work (Lines 71~101). Second, we provided more details of the I/O frameworks, including implementation method and future improvement plan in the discussion section (Lines 618~624). Lastly, we added several paragraphs to present our experiences and opinion about the pros and cons of our approach in the discussion section. (Lines 572~606)

Unfortunately, I did not succeed to install OpenArray on an OSX system (macOS 10.14.3, Armadillo 9.2, Boost 1.66, LLVM 7.0, gcc/g++/gfortran 8.3, openmpi 3.0).
**[Response]:**
Sorry to hear that. To solve the installation issue, we prepared a simple user manual for OpenArray v1.0, which is available at https://github.com/hxmhuang/OpenArray_CXX/tree/master/doc. In section 2 of the user manual, we introduced how to install OpenArray on Linux and Mac OS operating systems step by step. If you still have any question, please feel free to contact me (hxm@tsinghua.edu.cn).

**1.1.1 Introduction**

While the general motivation to start the OpenArray approach is made clear in this paper I am missing a more complete discussion of the state of the art. A few approaches (ATMOL, ICON DSL, STELLA) are listed but the text does not provide useful hints in how far OpenArray really goes beyond existing approaches. I am missing ATLAS (DOI: 10.1016/j.jcp.2017.07.006). I am not an expert in this field, but to me ATLAS seems to cover several design aspects, in particular operators, support for parallelism, and support for different grid types, and seems to be in these aspects similar to OpenArray. The ESCAPE project and its follow-up ESCAPE2 worked or will work in this direction. The authors cite Lawrence et al., 2018 but only as a reference for a trend towards the usage of "heterogeneous and advanced computing platforms", even though Lawrence et al. also discuss software approaches to address these challenges, including concepts like those used by OpenArray.

**[Response]:**

Thanks for your nice suggestions. We rewritten the introduction part, introducing more details about the related work, including ATMOL, ICON DSL, STELLA and ATLAS. In addition, we compared OpenArray with them in detail to demonstrate the advantages and disadvantages of OpenArray in the revised manuscript. (Lines 71~101)

"Many efforts have been made to address the complexity of parallel programming for numerical simulations, such as operator overloading, source-to-source translator and domain specific language (DSL). Operator overloading supports the customized data type and provides simple operators and function interfaces to implement the model algorithm. This technique is widely used because the implementation is straightforward and easy to understand (Corliss and Griewank, 1994; Walther et al., 2003). However, it is prone to work inefficiently because overloading execution induces numerous unnecessary intermediate variables, consuming valuable memory bandwidth resources. Using a source-to-source translator offers another solution. As indicated by the name, this method converts one language, which is usually strictly constrained by self-defined rules, to another (Bae et al., 2013; Lidman et al., 2012). It requires tremendous work to develop and maintain a robust source-to-source compiler. Furthermore, DSLs can provide high-level abstraction interfaces that use mathematical notations similar to those used by domain scientists, so that they can write much more concise and more straightforward code. Some outstanding DSLs, such as ATMOL (van Engelen, 2001), ICON DSL (Torres et al., 2013), STELLA (Gysi et al., 2015) and ATLAS (Deconinck et al., 2017), are used by the numerical model community. Although they seem source-to-source technique, DSLs are newly-defined languages and produce executable programs instead of target languages. Therefore the new syntax makes it difficult for the modellers to master the DSLs. In addition, most DSLs are not yet supported by robust compilers due to their relatively short history. Most of the source-to-source translators and DSLs still do not support the rapidly evolving heterogeneous computing platforms, such as the Chinese Sunway TaihuLight supercomputer which is based on the homegrown Sunway heterogeneous many-core processors and located at the National Supercomputing Center in Wuxi.

Other methods such as COARRAY Fortran and CPP templates provide alternative ways. Using COARRAY Fortran, a modeller has to control the reading and writing operation of each image (Mellor-Crummey et al., 2009). In a sense, one has to manipulate the images in parallel instead of writing serial code. In term of CPP templates, it is usually suitable for small code and difficult for debugging (Porkoláb et al., 2007)."

**1.1.2 IO**

As a model without IO is pretty useless it would have been nice to have read a few lines about how the (parallel) IO is approached. It is included in OpenArray, so why not spending one paragraph on such an important issues as well, perhaps with some graph showing the IO performance.

**[Response]:**
Thanks for your valuable suggestions. We added one paragraph in the discussion section to describe the current I/O interfaces,the implementing methods, and the future plan to improve the I/O performance of OpenArray. (Lines 618~624)

"Second, the data Input/Output is becoming a bottleneck of earth system models as the resolution increases rapidly. At present we encapsulate the PnetCDF library to provide simple I/O interfaces, such as load operation and store operation. A climate fast input/output (CFIO) library (Huang et.al, 2014) will be implemented into OpenArray in the next few years. The performance of CFIO is approximately 220% faster than PnetCDF because of the overlapping of I/O and computing. CFIO will be merged into the future version of OpenArray and the performance is expected to be further improved."

**1.1.3 Discussion**

You are convincing in the sense that your approach is valid and takes major burden from the oceanographer who "only" wants to run and modify an ocean model. On the other hand the complexity has not magically disappeared but it is moved from GOMO (in this example) to OpenArray. When porting the whole software onto a new system the major effort now goes into OpenArray - which is fine, but it has to be done. How complex is this? How flexible is the OpenArray approach in this respect.
**[Response]:**
Thanks for your helpful comments. Indeed, we moved the complexity from GOMO to OpenArray. Thus, the major burden of code porting is taken away for the oceanographers. OpenArray is designed to support multiple hardware platforms through separating hardware-dependent functions such as low-level numerical computations from the main framework.

We added a paragraph to introduce the complexity of the migration in section 3.4 (Lines 412-426): "With the help of dynamic code generation and JIT compilation technology, OpenArray can be migrated to different backend platforms. Several basic libraries, including Boost C++ libraries and Armadillo library, are required. The JIT compilation module is based on Low-Level-Virtual-Machine (LLVM), thus theoretically the module can only be ported to platforms supporting LLVM. If LLVM is not supported, as on the Sunway platform, one can generate the fusion kernels in advance by running the ocean model on an X86 platform. If the target platform is CPUs with acceleration cards, such as GPU clusters, it is necessary to add control statements in the CPU code, including data transmission, calculation, synchronous and asynchronous statements. In addition, the accelerating solution should involve the selection of the best parameters, for example "blockDim" and "gridDim" on GPU platforms. In short, the code generation module of OpenArray also needs to be refactored to be able to generate codes for different backend platforms. The application based on OpenArray can then be migrated seamlessly to the target platform. Currently, we have designed the corresponding source code generation module for Intel CPU and Sunway processors in OpenArray."

If the unlucky oceanographer comes up with the idea to try out yet another (perhaps higher-order) advection or any other scheme which is not yet supported by OpenArray, how difficult is it to extend OpenArray? Does this require expert knowledge and support from OpenArray developers? How seamless is - in your opinion - the integration of other stencils into the OpenArray library? I am not insisting on answering these questions line by line but rather take these as suggestions for what could be addressed in a thorough discussion. You as authors may wish to stress different - and in your opinion more important - points.

**[Response]:**
Thanks for your comments. We answered your questions about customized stencil operators in the discussion section. (Lines 599~606)

"The second issue is that the current OpenArray version cannot support customized operators. When modellers try out another higher-order advection or any other numerical scheme, the twelve basic operators provided by OpenArray are not abundant. We consider using a template mechanism to support the customized operators. The rules of operations are defined in a template file, where the calculation form of each customized operator is described by a regular expression. If users want to add a customized operator, they only need to append a regular expression into the template file."

**1.2 Specific Comments**

Line 39 and elsewhere: consider to replace "climate model" by "Earth system model", as the latter is now mainly used when talking about multi-component models in the context of Earth system and climate modelling efforts.
**[Response]:**
Thanks for the suggestion. In the revised manuscript, we have replaced "climate model" by "earth system model" in Line 35, 43, 58, 60, and 64.

Line 41: Please rephrase the sentence, as computing platforms are not applied but used.
**[Response]:**
Corrected. We have replaced "applied" with "used" in the sentence. (Line 46)

Line 55: Which gap do you precisely have in mind? Do you really mean climate science in general or rather climate modelling (aka Earth system modelling)?
**[Response]:**
Sorry for the confusion. The gap refers to a big obstacle between scientific inspiration and code implementation in the climate modeling community.

Therefore, we changed the sentence "… create a very large gap in climate science" with "…create a very large gap between scientific aspiration and code implementation in the climate modeling community". (Lines 60~62)

Line 70: What is the former, what is the latter language? Could you briefly explain to the non-experts amongst the readers the difference between source-to-source and DSL? Perhaps this whole paragraph needs some restructuring (see remarks in my section 1.1).
**[Response]:**
Thanks for your comments. In the sentence, the former and the latter stand for one language and another. As indicated by the name, source-to-source method converts one language, which is usually strictly constrained by self-defined rules, to another. DSLs are newly-defined languages and produce executable programs instead of target languages. As described in the section 1.1.1, we rewritten the paragraph to give a clear description (Lines 71~95).

Line 90: Please introduce the reader to the heterogeneity you have in mind here. What makes TaihuLight different in terms of heterogeneity? From the system specification further down in your text, TaihuLight does not look that heterogeneous.
**[Response]:**
Sorry for the confusion. Heterogeneity refers to systems deploying multiple types of processing elements within a single workflow, and allowing each to perform the tasks to which it is best suited (Shan Amar, 2006). The major innovation of the Sunway TaihuLight supercomputer is the homegrown Sunway many-core processor which consists of 4 core-groups. Each core-group includes 64 computing processing elements (CPEs) and a management processing element (MPE) (Fig. 1). CPE and MPE are different processing elements, the CPEs perform large-scale computing tasks and MPE are responsible for the task scheduling and communication. "The MPE is like a CPU core, and the CPE cluster is like a many-core accelerator, both the CPU and accelerator are now fused into one Sunway processor with a unified memory space" (Haohuan Fu et al, 2016, https://link.springer.com/article/10.1007/s11432-016-5588-7). The MPE-CPE hybrid architecture of Sunway TaihuLight makes Sunway TaihuLight heterogeneous and powerful.

[Figure]

Figure 1. The MPE-CPE hybrid architecture of the Sunway processor. Every Sunway processor includes 4 Core-groups (CGs) connected by the Network on Chip (NoC). Each CG consists of a management processing element (MPE), 64 computing processing elements (CPEs) and a memory controller (MC). The Sunway processor uses the system interface (SI) to connect with outside devices.

To make a clear description, we added the following sentences in the revised manuscript.

"Over the next few decades, tremendous computing capacities will be accompanied by more heterogeneous architectures which are equipped with two or more kinds of cores or processing elements (Shan, 2006), …" (Lines 54~56)

"…, such as the Chinese Sunway TaihuLight supercomputer which is based on the homegrown Sunway heterogeneous many-core processors and located at the National Supercomputing Center in Wuxi." (Lines 93~95)

"According to the TOP500 list released in November 2018, the Sunway TaihuLight is ranked third in the world, with a LINPACK benchmark rating of 93 Petaflops provided by Sunway many-core processors (or Sunway CPUs). As shown in Fig. 9, every Sunway CPU includes 260 processing elements (or cores) that are divided into 4 core-groups. Each core-group consists of 64 computing processing elements (CPEs) and a management processing element (MPE) (Qiao et al., 2017). CPEs handle large-scale computing tasks and MPE is responsible for the task scheduling and communication. The relationship between MPE and CPE is like that between CPU and many-core accelerator, except for they are fused into a single Sunway processor sharing a unified memory space." (Lines 428~437)

Line 100: I would say that you solved the problem for one ocean model or a particular class of ocean models but not yet for ocean models in general.
**[Response]:**
Thanks for your corrections. We have added "a particular class of" in front of "ocean models using the finite difference method and staggered grid in OpenArray." (Line 110)

Line 109: Is it really valid Fortran code, or shouldn't it better be classified as pseudo Fortran code as GOMO cannot really live without the OpenArray library?
**[Response]:**
Sorry for the confusion. OpenArray works as an independent library and it is the essential component of GOMO, since GOMO uses the functions and modules provided by OpenArray, such as average operators, differential operators, assignment functions, I/O functions, et al. Whereas, GOMO is written by valid Fortran codes.

Line 111: Is it really meant like that you compile and link one executable which can then be executed on any computing platform? Or are you talking about the intermediate C++ code? But this would require compiling and linking on the target system before it can be executed.
**[Response]:**
Sorry for the confusion. We transfer the cross-platform complexity to OpenArray. When porting the ocean models to a new platform, we need to redesign an additional function for the target system and add it to the code generation module in OpenArray. This added function is used to translate the intermediate computation graph into corresponding executable code for the target platform. At present, OpenArray supports X86 and Sunway platforms, therefore GOMO is executable on the two platforms without additional modification.

We rewritten the sentences (Lines 117~121): "Currently OpenArray supports both CPU and Sunway platforms. More platforms including GPU will be supported in the future. The complexity of cross-platform migration is moved from the models to OpenArray. The applications based on OpenArray can then be migrated seamlessly to the supported platform."

Line 120: True but only if the OpenArray has been ported to and is available on the Sunway platform. There is probably no free lunch when moving to a new hardware or software environment.

**[Response]:**
Agree. If we want to move GOMO to a new hardware or software environment, we should make OpenArray available on the new hardware platform since OpenArray is the footstone of GOMO. In fact, we transfer the burden of porting code from application to developing library, thus decoupling the ocean models from the hardware platforms is possible.

Therefore, we added the sentences to stress that (Lines 118~121): "The complexity of cross-platform migration is moved from the models to OpenArray. The applications based on OpenArray can then be migrated seamlessly to the supported platform."

Line 148 and elsewhere: Equation 1 is probably taken from the POM user manual, but nevertheless should not expressions like $\frac{\partial DU}{\partial x}$ rather be written as $\frac{\partial}{\partial x}(DU)$ ?

**[Response]:**
Thanks for your suggestions. Indeed, the Eq. 1 is derived from the POM user manual. Therefore, we would like to use the same expressions of the equations with that in POM user manual (shown as the following equation). This form is helpful to remove numbers of parentheses in Appendix A:Continuous governing equations.

$$\frac{\partial DU}{\partial x} + \frac{\partial DV}{\partial y} + \frac{\partial \omega}{\partial \sigma} + \frac{\partial \eta}{\partial t} = 0$$

Line 152: Could provide some hint on how to arrive at the discrete expression (2).

**[Response]:**
Thanks for your suggestions. We added the following details of Arakawa C grid and finite difference method to demonstrate the process from *Eq. (1)* to the discrete *Eq. (2).* (Lines 161~172)

"In Arakawa C grid, *D* is calculated at the centers, *U* component is calculated at the left and right side of the variable *D*, *V* component is calculated at the lower and upper side of the variable *D* (Fig. 1). Variables (*D, U, V)* located at different positions own different sets of gird increments. Taking the term $\frac{\partial DU}{\partial x}$ as an example, we firstly apply linear interpolation to obtain the *D's* value at *U* point represented by *tmpD*. Through a backward difference to the product of *tmpD* and *U*, then the discrete expression of $\frac{\partial DU}{\partial x}$ can be obtained.

    *tmpD(i+1,j) = 0.5\*(D(i+1,j)+D(i,j)),*       (2)
and

$$\frac{\partial DU}{\partial x} = \frac{tmpD(i+1,j) - tmpD(i,j)}{dx(i,j)^*} = \frac{0.5*\big(D(i+1,j)+D(i,j)\big)*U(i+1,j) - 0.5*\big(D(i,j)+D(i-1,j)\big)*U(i,j)}{dx(i,j)^*},$$

(3)

where $dx(i,j)^* = 0.5*(\,dx(i,j) + dx(i-1,j)\,).$ ''

[Figure]

Figure 2. Arrangement of variables in the staggered Arakawa C grid. (Numbered as 'Figure 1' in the revised manuscript)

Line 211: I thought your current implementation of the operators only supports uniform (=equidistant) grids? What am I missing here?
**[Response]:**
Sorry for the confusion. In OpenArray, grid increments ($dx$, $dy$, $dz$) are combined with physical variables through grid binding. On the staggering Arakawa C-grid, the components of velocity ($u$, $v$, $w$) and potential temperature ($T$) et al are defined at different points. Variables on each point has its own set of grid increments so the current implementation of the operators supports varying grids.

We have changed the sentence "…, the operators can automatically set the correct grid increments for different Array variables." into "…, a set of grid increments, including horizontal increments ($dx(i,j)$, $dy(i,j)$) and vertical increment ($dz(k)$), will be combined with each corresponding physical variable through grid binding. Thus, the operators can implicitly set the correct grid increments for different *Array* variables, even if the grid is nonuniform." (Line 236~240)

Line 214 and elsewhere: I suggest to avoid the phrase "automatic". Nothing is done automatically but every effect has a cause. Here, something is happening because you programmed it that way and some conditions are coming together to trigger an action.
**[Response]:**
Thanks for your corrections. We have replaced all the phrases "automatic/automatically" with "implicit/implicitly" in the revised manuscript. A programmer that writes implicitly parallel code does not need to worry about task division or process communication, focusing instead on the problem that his or her program is intended to solve.[From wikipedia: Implicit parallelism]

Line 238: Could you clarify how the Arakawa grid type, the jumping rules and the differential operators are linked together? Let us assume I have formulated my ocean model on an Arakawa C grid and now for curiosity would like to run it on an A grid (not because it would really makes sense but to demonstrate the effect of discretization on the numerical solution) what would I have to change in my ocean model code?
**[Response]:**
Thanks for your valuable suggestion. To make it clearer, we took the Eq. (1) switching from Arakawa C grid to Arakawa B grid as an example, to illustrate the modifications to the ocean models when grid scheme is changed in the revised manuscript. (Lines 264~274)

"If users change the Arakawa grid type, first the position information of each physical variable need to be reset (Shown in Fig. 4). Then the discrete form of each equation needs to be redesigned. We take the Eq. (1) switching from Arakawa C grid to Arakawa B grid as an example. The positions of the horizontal velocity Array U and Array V are changed to Point 0, Array η and Array D stay the same. The discrete form is changed from Eq. (4) to Eq. (13), the corresponding implementation by operators is changed from Eq. (6) to Eq. (14)."

$$\frac{\eta_{t+1}(i,j)-\eta_{t-1}(i,j)}{2*dt} + \frac{0.25*(D(i+1,j)+D(i,j))*(U(i+1,j)+U(i+1,j+1))-0.25*(D(i,j)+D(i-1,j))*(U(i,j)+U(i,j+1))}{dx(i,j)^*} +$$

$$\frac{0.25*(D(i,j+1)+D(i,j))*(V(i,j+1)+V(i+1,j+1))-0.25*(D(i,j)+D(i,j-1))*(V(i,j)+V(i+1,j))}{dy(i,j)^*} = 0$$

$$(13)$$

$$\eta_{t+1} = \eta_{t-1} - 2 * dt * \left( \delta_f^x \left( \overline{D}_b^x * \overline{U}_f^y \right) + \delta_f^y \left( \overline{D}_b^y * \overline{V}_f^x \right) \right) \tag{14}$$

Line 247: What is the motivation for the list of ocean model codes you provide in this paragraph? There are other codes around, e.g. FESOM (see https://fesom.de and the list of publication there) or an unstructured grid model for global ocean dynamics by P. Korn (see https://doi.org/10.1016/j.jcp.2017.03.009) and several more.
**[Response]:**
Sorry for the confusion. In this paragraph, we want to express our respect to these excellent ocean models, such as POM, ROMS, MITgcm, and FESOM. From these models, we obtained abundant experience and knowledge leading us to build OpenArray. Considering most of these existing ocean models using finite difference or finite volume methods on structured meshes, we designed 12 basic operators in OpenArray only for this particular class of ocean models at present. In our future work, more customized operators will be implemented to support other numerical methods and meshes.

We simplified the paragraph (Lines 284~291): "Although most of the existing ocean models use finite difference or finite volume methods on structured or semi-structured meshes (e.g., Blumberg and Mellor, 1987; Shchepetkin and McWilliams, 2005), there are still some ocean models using unstructured meshes (e.g., Chen et al., 2003; Korn, 2017), and even the spectral element method (e.g., Levin et al., 2000). In our current work, we design the basic operator only for finite difference and finite volume methods with structured grids. More customized operators for the other numerical methods and meshes will be implemented in our future work."

Line 274 section 3.1: Could you add a few lines to describe the handling of lateral and vertical boundary conditions within your operators?
**[Response]:**
Thanks for your comments. In section 3.3, we added the details to describe how to handle the lateral and vertical boundary conditions within the operators (Lines 406~409): "For the global boundary conditions of the limited physical domains, the values at the physical border are always set to zero within the operators and operator expressions. In realistic cases, the global boundary conditions are set by a series of functions (e.g., radiation, wall) provided by OpenArray."

Line 334: When speaking of subgraphs and the kernel function, can the individual advection and diffusion terms be accessed for diagnostic purposes?
**[Response]:**
Thanks for your comments. If users want access to individual variables or subgraphs, they need to split the formula or code into multiple expressions for diagnostic or printing purposes.

We added the sentence to introduce the access to any subgraphs (Lines 367~369): "Users can access to any individual subgraph by assigning the subgraph to an intermediate variable for diagnostic purposes."

Line 352: I am not sure what is meant here and perhaps the sentence should be rephrased. Such a function needs to be programmed once for a particular ocean model, but once it is there it can be used, see e.g. ESMF_FieldBundleHalo contained in the Earth System Modeling Framework (ESMF). To my knowledge, other ocean models use similar approaches for the halo exchange as well. But no doubt, it is a relief to have it.
**[Response]:**
Thanks for your suggestions. Earth system models generally provide simple functions (e.g. ESMF_FieldBundleHalo) for users to manually control the communication of boundary regions. In OpenArray, we further hide these communication through the fused kernel. Users do not need to care about the halo exchange, or know the parallel details.

The sentence has been rephrased (Lines 396~397): "Clearly, ocean modellers have to frequently call corresponding functions to carefully control the communication of the local boundary region."

Line 391: Is the mode splitting algorithm inherited from POM, if so, this should be mentioned.
**[Response]:**
Thanks for your suggestion. Indeed, the mode splitting algorithm is inherited from POM. Therefore, we have changed "… the mode-splitting algorithm to address …" into "… the mode-splitting algorithm inherited from POM to address …" (Lines 454~455)

Line 422: Could you provide the number of lines for OpenArray as well? I could calculate it myself but . . .
**[Response]:**
Thanks for your suggestion. The code of OpenArray is about 11,800 lines. We added the sentences to provide the number of lines for OpenArray. (Lines 493~495): "since the complexity has been transferred to OpenArray, which includes about 11,800 lines of codes."

Line 425: You raise the impression that porting is not an issue anymore. While this is certainly true for GOMO (which is of course very valuable) the porting still has to be done for OpenArray. Maybe this could be clarified somewhere (perhaps in the discussion).
**[Response]:**
Thanks for your comments. Indeed, since the porting has been transferred to OpenArray, the models can be run on these platforms supported by OpenArray.

Therefore, we added the clarification in our revised manuscript. (Lines 118~121): "The complexity of cross-platform migration is moved from the models to OpenArray. The applications based on OpenArray can then be migrated seamlessly to the supported platform."

Line 469 section 5.3: Which hardware do you use for these tests? What sets the upper bound of 4096 processes?
**[Response]:**
Thanks for your comments. In section 5.3, the strong and weak scaling experiments were taken on the X86 cluster at National Supercomputing Center in Wuxi of China, which provides 5000 Intel Xeon E5-2650 v2 CPUs for our account at most. In the future, we will increase the upper bound if more computing resources are available.

We added the sentences to provide more details (Lines 543~545): "We use the X86 cluster at National Supercomputing Center in Wuxi of China, which provides 5000 Intel Xeon E5-2650 v2 CPUs for our account at most."

Line 481 section 5.4: What does this mean for a reasonable local domain size? 32X32 points as in sec. 5.3 is still on the good side, while 9x9 as used on TaihuLight does shows some performance degradation.

**[Response]:**

Thanks for your comments. On the same machine architecture, the parallel efficiency is usually better if the local domain size is bigger. The weak and strong scaling experiments in section 5.3 were taken on X86 cluster, while the experiment in section 5.4 was taken on the Sunway processors. As described above, the architecture of Sunway platform is largely different from X86 machine.

We added the sentences in section 5.3 to make a clear description (Lines 543~545): "We use the X86 cluster at National Supercomputing Center in Wuxi of China, which provides 5000 Intel Xeon E5-2650 v2 CPUs for our account at most."

In section 5.4, we changed the sentence "The strong scalability of GOMO is also tested on the Sunway TaihuLight supercomputer." into "We also test the scalability of GOMO on the Sunway platform." (Line 555)

Line 490: As I understand the steps up to compiling the JIT are done only once at the beginning of a job. If you run a longer experiment (in terms of wallclock time or number of timesteps) the initialisation phase should be negligible when compared to the total run time. Why don't you provide two numbers, one for the initialisation and one for the integration within the time loop?

**[Response]:**

Agree. The fusion-kernel codes are generated and compiled only once at the beginning of a job. In the scalability tests of GOMO, this initialization phase consumes about 2 minutes.

We added the sentence to introduce the time consumption (Lines 564~566): "Even though the fusion-kernel codes are generated and compiled only once at the beginning of a job, it consumes about 2 minutes."

**1.3 Technical Corrections/Suggestions**

L39 climate model → climate models

**[Response]:**

We changed the "climate model" into "earth system models". (Line 43)

L42 climate community → climate modelling community

**[Response]:**

Corrected.

L43 model program needs → model programs need

**[Response]:**

Corrected.

L68 inefficiency → inefficiently
**[Response]:**
Corrected.

L83 the sentence needs to be reordered. It is not clear (to me) to which part "at the product level" is referring to.
**[Response]:**
Sorry for the confusion. We rewritten the whole paragraph. To make a clear description, the sentence has been replaced with "It requires tremendous work to develop and maintain a robust source-to-source compiler." (Lines 81~82)

L106 change to : . . . is similar to the original but manually optimized parallel program.
**[Response]:**
Corrected.

L108 support → supports
**[Response]:**
Corrected.

L125 → The implementation
**[Response]:**
Corrected.

L139 → In traditional ocean models . . .
**[Response]:**
Corrected.

L143 → When using the OpenArray library . . .
**[Response]:**
Corrected.

L211 → we propose
**[Response]:**
Corrected.

L220 → . . . the horizontal velocity components Array(U) and Array(V) are . . .   or
          . . . the horizontal velocity Array(U, V) is . . .
**[Response]:**
Thanks for your corrections. We have changed the sentence ". . . the horizontal velocity Array(U, V) are . . ." into ". . . the horizontal velocity *Array* (*U, V*) is . . ." (Line 246)

L238 can be used → are used

**[Response]:**
Corrected.

L257 different → difference; operator → operators
**[Response]:**
Corrected.

L289 will be concealed by → is hidden behind
**[Response]:**
Corrected. We changed "will be concealed by" into "are hidden behind".

L290 → : : : and can escape . . .
**[Response]:**
Corrected.

L303 → to implement a so-called lazy expression . . .
**[Response]:**
Corrected.

L318 . . .AYF are the interpolated functions → . . .AYF are the average functions.
**[Response]:**
Corrected.

L373 computing processing elements: aren't these Central Processing Units (CPUs)
**[Response]:**
Sorry for the confusion. The computing power of TaihuLight is provided by a homegrown Sunway many-core processor (or Sunway CPU). The Sunway CPU includes four core-groups (CGs). Each CG includes one management processing element (MPE), 64 computing processing elements (CPEs), and one memory controller (MC). CPEs can only perform computations and MPE are responsible for the task scheduling and communication. Traditionally, the term "CPU" refers to a processor, including processing unit and control unit at least. Therefore, computing processing elements are not regarded as CPUs.

Therefore, we added more details of computing processing elements in section 3.4 (Lines 433~437): "CPEs handle large-scale computing tasks and MPE is responsible for the task scheduling and communication. The relationship between MPE and CPE is like that between CPU and many-core accelerator, except for they are fused into a single Sunway processor sharing a unified memory space."

L384 a practical ocean model → a numerical ocean model
**[Response]:**
Corrected.

L404 rephrase, the TKE and alike can be calculated but not the submodel.

**[Response]:**

Thanks for the suggestion. We have changed the sentence "…, and turbulence closure sub-model (q2, q2l) (Mellor and Yamada, 1982)" into "…, and turbulence closure scheme (q2, q2l) (Mellor and Yamada, 1982)".

L505 a practical ocean model → a numerical ocean model

**[Response]:**

Corrected.

**2 Responses to the comments of David Webb**

Thanks for your valuable feedbacks again. Your comments for the OpenArray and my previous article (POM.gpu) are always highly insightful and enable us to improve the quality of our manuscripts. Below are our point-by-point responses to all your comments and our plans to revise the manuscript.

1. "This is a well written paper concerned with generating an ocean model from a set of equations closer in form to the underlying differential equations than is usual. It is an interesting computational exercise but the resulting code has some important deficiencies and for that reason I think it would be more suited for a computational journal than the present one."

**[Response]:**

We are very happy to hear that you are satisfied with our paper written. Although it looks like a general computing tool, OpenArray is particularly designed for earth system models. It is a product of collaboration between oceanographers and computer scientists. We aim to promote it to the model community as a development tool for the future numerical models. We believe it is of great help for the development of ocean models and we sincerely hope it to be widely used by ocean modellers. That is why we submit our manuscript to GMD to advertise our work.

We have modified the abstract (Lines 23~26) to stress its significance for the model development, extended the introduction (Lines 71~101) to compare with existing studies, and added a new discussion section (Lines 572~635) to describe the pros and cons of OpenArray.

2. "My main concerns all involve computational efficiency. As the authors state, ocean models are memory bandwidth limited and for that reason the code is usually written in such a way that once a variable is in one of the processor registers or in the highest speed cache it is used as much as possible before being replaced. In fact the aim should be never to move variables more than once each timestep. In the present code the authors spend a major section reporting on one small step in this direction, but really this should only be the first of many steps."

**[Response]:**

Totally agree, computational efficiency is a big challenge. We have to admit we cannot fully solve the memory bandwidth limited problem at the current stage, but the main purpose of OpenArray is to provide a user-friendly, platform-independent tool for the development of ocean models. The memory consumption by similar techniques such as operator overloading is much higher due to the unnecessary intermediate variables. In contrast, the memory consumption by OpenArray is significantly reduced to a level similar to the ocean models developed in the conventional way. In OpenArray, we adopted a series of optimization methods to alleviate the requirement of memory bandwidth. We will further make continuous efforts to optimize the performance of OpenArray in the future version, using the techniques including time skewing and polyhedral model. Therefore, we added the following paragraphs to discuss the memory bandwidth limitation. (Lines 583~597)

"However, there are still several problems to be solved in the development of OpenArray. The first issue is computational efficiency. Once a variable is in one of the processor registers or in the highest speed cache, it should be used as much as possible before being replaced. In fact, we should never to move variables more than once each timestep. The memory consumption brought by overloading techniques is usually high due to the unnecessary variable moving and unavoidable cache missing. The current efficiency and scalability of GOMO are close to sbPOM, since we have adopted a series of optimization methods, such as memory pool, graph computing, JIT compilation, and vectorization, to alleviate the requirement of memory bandwidth. However, we have to admit that we cannot fully solve the memory bandwidth limited problem at present. We think that time skewing is a cache oblivious algorithm for stencil computations (Frigo and Strumpen, 2005), since it can exploit temporal locality optimally throughout the entire memory hierarchy. In addition, the polyhedral model may be another potential approach, which uses an abstract mathematical representation based on integer polyhedral, to analyze and optimize the memory access pattern of a program."

3. "I am also concerned about the way the code deals with multi-processor architectures. Unfortunately, although I was able to compile both the c++ and fortran sections, the link step failed and so I was not able to check the running code. However the main time stepping loop appears to run on a single processor and it is only the difference and averaging operators, in the c++ code, which make use of the multi-processor architecture. This is surprising given the authors emphasis on the importance of parallel computing."

**[Response]:**
We are so sorry for the failure in the link step. The other referee also reported this bug. We have fixed this bug and we submitted a simple user manual including installation instructions, description of functions, application examples and debug methods, etc. on the GitHub (https://github.com/hxmhuang/OpenArray_CXX/tree/master/doc).

We emphasis on the importance of parallel computing because the operators in OpenArray, not only the difference and average operators, but also the "+", "-", "*", "/" and "=" operators in the Fortran code, are all implicit parallelism. It is the most prominent feature of the program using OpenArray.

Therefore, we added a paragraph in Section 4 to describe more details (Lines 473~479):
"When the user dives into the GOMO code, the main time stepping loop in GOMO appears to run on a single processor. In fact, implicit parallelism is the most prominent feature of the program using OpenArray. The operators in OpenArray, not only the difference and average operators, but also the "+", "-", "*", "/" and "=" operators in the Fortran code, are all overloaded for the special data structure "Array". The seemly serial Fortran code is implicitly converted to parallel C++ code by OpenArray, and the parallelization is hidden from the modellers."

4. "I am also a bit wary about all the details being lost in the c++ compiler/interpreter code. The authors emphasis the possibility of portability but this implies a large organization continually keeping such a code up to date and adapted to the latest hardware. If not, the climate modelling groups have to do it themselves in which case the effort required will be much the same as now except for the addition of the compiler/interpreter and associated packages."

**[Response]:**

Thanks for your valuable suggestions. Indeed, we aim to develop OpenArray into a stable and community software, just like the C/C++/Fortran compiler or math library. We will follow the GNU open source license. In general, the programmers do not need to worry about the technical details of a compiler too much, especially for the oceanographers. Our original intention was to provide simple Fortran interfaces and the C++ section is totally hidden by OpenArray. In addition, the developing team of OpenArray consists of researchers from the National Supercomputing Center in Wuxi and Tsinghua University. OpenArray will be an important software to simplify the porting work on the Sunway TaihuLight supercomputer. The Sunway TaihuLight supercomputer has been the world's fastest supercomputer for two years, from June 2016 to June 2018, according to the TOP500 lists. The OpenArray project is supported by some long-term grants. Therefore, the project will be stably maintained for at least 4 years.

We added some sentences in the section 3.3 to introduce how the subgraphs are converted into C++ code (Lines 374~379): "In order to generate a kernel function based on a subgraph, we firstly add the function header and variable definitions according to the name and type in the *Array* structure. And then we add the loop head through the dimension information. Finally, we perform a depth-first walk on the expression tree to convert data, operator, and assignment nodes into a complete expression including load variables, arithmetic operation, and equal symbol with C++ language."

In addition, we added the following paragraph to stress the aim of OpenArray and its role in model development (Line 631~635): "OpenArray is a product of collaboration between oceanographers and computer scientists. It plays an important role to simplify the porting work on the Sunway TaihuLight supercomputer. We believe that OpenArray and GOMO will continue to be maintained and upgraded. We aim to promote it to the model community as a development tool for the future numerical models."

5. "Both of the architectures discussed appear to be cache based but as I understand it the next major advance will come from better use of gpus. In these systems I expect variables will stay in gpu memory or be swapped between gpus and rarely return to the main cpu memory. For such systems the structure proposed here appears unsuitable."

**[Response]:**

Thanks for your helpful comments. In addition to CPU platform, the current version of OpenArray supports the Sunway TaihuLight supercomputer which represents the major advance in China, since we are affiliated to the National Supercomputing Center in Wuxi. As you know, most of the current ocean models are still running on the CPU architecture. Our plan is to develop stable and efficient OpenArray on the both discussed architectures first, and then migrate it to GPU. Actually, the GPU version of OpenArray is already under development.

We introduced the details of GPU version of OpenArray in the discussion section. (Lines 611~616): "First, we are developing the GPU version of OpenArray. During the development, the principle is to keep hot data staying in GPU memory or directly swapping between GPUs and avoid returning data to the main CPU memory. NVLink provides high bandwidth and outstanding scalability for GPU-to-CPU or GPU-to-GPU communication, and addresses the interconnect issue for multi-GPU and multi-GPU/CPU systems."

6. "There is also the question of how researchers might try out new code with the prosed library and debug the resulting runs. No user manual is provided and it is difficult to see how bugs can be traced, especially if they involve the c++ section of the code."

**[Response]:**

A user manual and several application examples have been released on GitHub to help users to install and use OpenArray, a reference manual including more details is currently under preparation. In addition, we have added several functions for debugging (shown in Tab. 1) and these functions are proved very useful during the ocean model (GOMO) development. In the future, more debugging functions will be implemented to trace and solve the potential bugs.

Table 1. Functions for debugging in OpenArray

| Functions | Description |
|---|---|
| display() | print the value of an array |
| display_array_info() | print the information of an array |
| save() | save an array to a given file |
| sum() | sum array in a given direction |
| csum() | cumulative sum an array in a given direction |
| max() | get the maximum value of an array |
| min() | get the minimum value of an array |
| max_at() | get the position of the maximum value |
| min_at() | get the position of the minimum value |

We added a sentence in the Code availability section (Lines 654~655): "… and the user manual of OpenArray can be accessed at https://github.com/hxmhuang/OpenArray_CXX/tree/master/doc".

7. "I also do not understand why a just-in-time compiler is used, given that the model grids do not change in time so that both human and computer effort would be better spent optimizing the fixed grid code. And on this theme I am also concerned, although I would like to be proved wrong, that what has been achieved here is little different from what might be achieved with a good fortran coarray code, together with statement functions or cpp define statements to take the place of the operators."
**[Response]:**
Thanks for your helpful comments. The just-in-time compiler used in OpenArray can fuse numbers of operators into a large compiled kernel. The benefit of fusing operators is to reduce memory bandwidth requirements and improve performance compared to executing operators one-at-a-time. Comparing with COARRAY Fortran, OpenArray supports implicit parallelism so that a modeller does not need to worry about task division or process communication. Using COARRAY Fortran, a modeller has to control the reading and writing operation of each image. In a sense, one has to manipulate the images in parallel instead of writing serial code. In term of CPP templates, it is usually suitable for small code and difficult for debugging.

Therefore, we added the sentences to explain why we used the JIT compiler (Lines 371~374): "The JIT compiler used in OpenArray can fuse numbers of operators into a large compiled kernel. The benefit of fusing operators is to alleviate memory bandwidth limitations and improve performance compared with executing operators one-by-one."

In addition, we added the difference between OpenArray and COARRAY Fortran/CPP templates (Lines 97~101): "Other methods such as COARRAY Fortran and CPP templates provide alternative ways. Using COARRAY Fortran, a modeller has to control the reading and writing operation of each image (Mellor-Crummey et al., 2009). In a sense, one has to manipulate the images in parallel instead of writing serial code. In term of CPP templates, it is usually suitable for small code and difficult for debugging (Porkoláb et al., 2007)."

8. "One reason that these are not used in ocean models comes from the fact that in a typical ocean model only about half the theoretical 3-D grid is involved in the calculation, the rest representing land or ocean topography. When computer memory and power is readily available a factor of two does not really matter but given the computational cost of many ocean models, spending time on such cells, as usually happens when coarrays are used, can be critical."
**[Response]:**
Thanks for your valuable comments. Indeed, the load imbalance is one of the major issues about computational efficiency of ocean models. The land points accounting for over 30 percent are needed to be masked in the ocean grid. It is a common problem in most of the existing ocean models. We will adopt the space-filling curve and curvilinear orthogonal grid method to solve this issue in our future version.

Therefore, we added a paragraph in discussion section to emphasize this computational problem and provide several possible solutions (Lines 626~629): "Finally, as most of the ocean models, GOMO also faces the load imbalance issue. We are adding the more effective load balance schemes, including space-filling curve (Dennis, 2007) and curvilinear orthogonal grids, into OpenArray in order to reduce the computational cost on land points."

9. "Anyway - you can see that I am unhappy. However I must emphasise that I can also see that the paper represents a lot of hard work and I accept that as an example of an attempt to tackle good computational problem it is worthwhile. For this reason I think that publication in a journal more closely linked to the development of artificial intelligence would be more suitable."
**[Response]:**
We sincerely appreciate your affirmation of our work. In fact, OpenArray is based on the joint efforts of climate modeling and high-performance computing scientists. We aim to promote it to the model community as a development tool for the future numerical models. Thus we think this work fits the aim and scope of GMD well.

We added the following sentences to introduce the role of OpenArray in model development (Lines 631~635): "OpenArray is a product of collaboration between oceanographers and computer scientists. ……, we aim to promote it to the model community as a development tool for the future numerical models."

10. "If on the basis of the other referees reports, the authors are asked to provide a revised version then there are two extra documents I would like to see in the additional documentation section. The first is a user manual. The second is a full list of the include files and libraries needed (i.e. all those which are not the fortran or c++ compiler files) - to help with the problems I had."
**[Response]:**
Thank you for the opportunity. We have finished a simple user manual. In the installation section, we listed all pre-installed software and libraries required by OpenArray. The user manual is available on GitHub now (https://github.com/hxmhuang/OpenArray_CXX/tree/master/doc).

We really appreciate your highly constructive comments. We hope you will be satisfied with the above reply. If there are any questions, please contact us for free.

Best wishes,
Xiaomeng Huang, Xing Huang, Dong Wang, and Yi Li.

[revised manuscript text omitted]

                          .

---

## Author Response (AR2)

Dear editor and reviewers,

First of all, we would like to express our sincere appreciation to your valuable feedbacks. Your comments are highly insightful and enable us to substantially improve the quality of our manuscript and OpenArray.

In the previous reviews, both reviewers and the editor suffered some issues compiling OpenArray. To ease the installation of OpenArray, we have split it into two packages, a release one for users (https://github.com/hxmhuang/OpenArray) and a developing one for project team members (https://github.com/hxmhuang/OpenArray_Dev). The release package includes all the final operator functions and does not require any external libraries (e.g., Python, Boost, Ctags, LLVM, JIT, and Git) other than gcc/g++/gfortran compilers, MPI library, and PnetCDF (shown in the Tab. 1). For the users who are interested in the details of how the C++ code is generated, the developing package can be referred to. The release package enables users to install OpenArray from source through three simple steps (./configure; make; make install). The configure file can detect the environment and generate corresponding Makefile, as many apps do. In addition, we have tested the release package on many platforms, including macOS and quite a few Linux distros (including different versions of Fedora, CentOS, Ubuntu, openSUSE and Debian). We believe the installation should be much easier and the source code is much more robust. The thorough installing guide is given in the updated user manual (https://github.com/hxmhuang/OpenArray/tree/master/doc).

Table 1 Dependency of the release package

| Type | Source package |
|------|----------------|
| Dependency | gcc/g++/gfortran compilers, version 4.9.0 or later. |
|  | MPI compilers (mpich v3.2.1 or later; openmpi v3.0.0 or later; Parallel Studio XE 2017 or later) |
|  | PnetCDF, version 1.7.0 or later. |
| Notes | In the release package, the function of Boost, JIT, LLVM, and Armadillo has been added into OpenArray. In addition, the pre-process which uses Python is removed. Git and Ctags is optional. |

Below are our point-to-point responses to all comments.

**1 Responses to Dr. Steven J. Phipps**

(1) I am afraid, therefore, that I agree with Referee #2 that OpenArray is not yet in a sufficiently robust state for me to allow publication in Geoscientific Model Development.

Ultimately, while your manuscript fulfils the GMD mission of documenting the software, it remains the case that the software described must also be useful to the community.

I have therefore made the decision "Reconsider after major revisions". Referee #1 has suggested that the text of the manuscript would benefit from proof-reading. Apart from this, I would like to emphasise that the text of the manuscript itself is satisfactory and you should not consider that you need to make changes.

Rather, my reason for returning the manuscript to you at this stage is to allow you to revise the manual and source code of OpenArray, in response to the comments of both referees and in response to my own comments. I hope you can agree that your manuscript will be strengthened considerably if OpenArray is a robust tool that can readily be adopted by the community.

**Response:**

We sincerely appreciate your efforts installing OpenArray and valuable comments. As described above, we have first minimized the dependencies on the many external libraries for OpenArray users (e.g., Python, Boost, Ctags, LLVM, JIT, and Git). Second we have rebuilt an easy-to-use release package, which enables users to install OpenArray from source by three simple steps (./configure; make; make install). The release package and scripts has been tested on different platforms (including MacOS, openSUSE, Fedora, Ubuntu, Debian, Centos) using different compilers with mpi libraries (e.g., GCC+mpich, GCC+openmpi, Intel Parallel Studio XE). We believe that OpenArray are stable and robust enough to be adopted by the modelling community.

We have revised the manual on GitHub (https://github.com/hxmhuang/OpenArray/tree/master/doc) and modified the corresponding sentence in the manuscript. (Lines 653-655):

In the revised manuscript, we changed the sentences "The source codes of OpenArray v1.0 is available at https://github.com/hxmhuang/OpenArray_CXX, and the user manual of OpenArray can be accessed at https://github.com/hxmhuang/OpenArray_CXX/tree/master/doc." into "The source codes of OpenArray v1.0 is available at https://github.com/hxmhuang/OpenArray, and the user manual of OpenArray can be accessed at https://github.com/hxmhuang/OpenArray/tree/master/doc."

In the revised user manual, we added a installing guide for the release version of OpenArray and GOMO. For more details, please refer to the user manual diff.

(2) On my machine, the names of the Intel MPI compilers are mpifort, mpicc and mpic++. The compiler flag to enable OpenMP is also -openmp, rather than -qopenmp. Both of these required me to edit the makefile.

**Response:**

Thanks for your helpful comments. Using different versions of Intel MPI compilers will vary the compilation instructions and compiler flags. In the Inter compilers version 2017 or later, the names of the Intel MPI compilers are mpiifort, mpiicc, and mpiicpc, and the compiler flag to enable OpenMP is -qopenmp. For the release package, the configure file can detect the environment and generate the Makefile automatically. Therefore, users do not need to modify the makefile.

(3) I had to install the ctags package to avoid an error at the precompile stage.
**Response:**
In the release package, the dependency of OpenArray on ctags is removed.

(4) However, I was unable to compile the test case. Firstly, it would be helpful if the instructions could make it clear that you need to be in the build/ directory, rather than the top-level directory. Nonetheless, the main problem is simply that the model does not compile:

*mpifort -O0 -w -g -std=c++0x -DBOOST_LOG_DYN_LINK -openmp -D_WITHOUT_LLVM_ -o manual_main user-manual/oa_main.o \*
*-lpnetcdf -lboost_program_options -lboost_filesystem -lboost_system -lboost_log -lboost_log_setup -lboost_thread -ldl -ljit -lgfortran -Wl,-rpath=/lib64/ -lstdc++ -L/lib64/ -lpnetcdf -lboost_program_options -lboost_filesystem -lboost_system -lboost_log -lboost_log_setup -lboost_thread -ldl -ljit -lgfortran -Wl,-rpath=/lib64/ -lgfortran -L. -lopenarray -lm -ldl -lstdc++ -lboost_program_options \*
*-lboost_system -lboost_log -lboost_log_setup -lboost_thread -lpnetcdf -lpnetcdf*

*/usr/bin/ld: cannot find -ljit*
*/usr/bin/ld: cannot find -ljit*
*collect2: error: ld returned 1 exit status*
*makefile.intel:146: recipe for target 'manual_main' failed*

Clearly, there is at least one undocumented dependency on an external library that is causing compilation to fail.
**Response:**
Thanks for your valuable suggestions. The previous installation instruction is misleading. In the revised version, there is no need to install the JIT library as well as some other apps, since the dependency has been removed.

**2 Responses to Dr. David Webb**

(1) After some effort (see below) I managed to compile the programs and routines in the top level and in directory 'c-interface'. The compile process then failed in the 'modules' directory:

==============================
mpicxx -O3 -ffast-math -DBOOST_LOG_DYN_LINK -fopenmp -c -fPIC --std=c++0x -Werror=return-type                -fno-trapping-math                -fno-signaling-nans modules/tree_tool/Simple_Node.cpp -o modules/tree_tool/Simple_Node.o
modules/tree_tool/Simple_Node.cpp: In member function 'void* Simple_node::get_val()':

modules/tree_tool/Simple_Node.cpp:30:1: error: control reaches end of non-void function [-Werror=return-type]
}
^
cc1plus: some warnings being treated as errors
make: *** [makefile.linux:63: modules/tree_tool/Simple_Node.o] Error 1
============================

**Response:**

Sorry for the failure in your compiling process. In the User Manual, we recommended installing OpenArray with the below instruction:

   *make -f makefile.intel oalib_obj.*

The makefile *makefile.intel,* rather than *makefile.linux*, should be used here. However, we have simplified the installing process by splitting OpenArray into release and developing packages. The release package does not require external libraries other than gcc/g++/gfortran compilers, MPI library, and PnetCDF. In addition, the release package is easy to use and has been tested on many platforms, including macOS and a few Linux distros (Fedora, CentOS, Ubuntu, openSUSE, and Debian). For more details, please refer to the revised user manual (https://github.com/hxmhuang/OpenArray/doc/).

(2) I was conscious last time of the number of additional packages that I had to install before I could compile the programs. One of the problems seemed to be that the authors were use to using scripting packages, for example fypp, which are not usually used by the community running geophysical models. At the same time they were treating gcc, fortran, openmpi and netcdf as being special.

**Response:**

Thanks for your suggestions. We have removed the dependency of such packages from the release version of OpenArray.

(3) Another oddity was the use of pnetcdf, somewhat outdated, and a lack of specificity about which version of python or openmpi to use.

**Response:**

Thanks for your suggestions. We used pnetcdf at this moment. In the future version of OpenArray, a climate fast input/output (CFIO) library (Huang et.al, 2014) will be implemented to achieve better efficiency. OpenArray requires Open MPI v3.0.0 or later, as specified in the revised user manual. Python is not needed anymore by the release version.

(4) This time I started with a clean install of opensuse 15.1, the KDE desktop version. The use manual talks of compiling the various subroutine libraries etc, but my experience last time and with other systems is that each library usually requires at least one other library to be compiled and the whole process can take for ever.

Instead modern distributions use packages, installation of one package automatically installing any additional required packages. The authors need to revise their user manual to reflect this.

With opensuse 15.1 I could install packages:

gcc7-7.4.0, gcc7-fortran-7.4.0, gcc7-c++-7.4.0 (+ 12 packages)
openmpi1-gnu-hpd-devel (+13 packages)
pnetcdf (+26 packages)
armadillo-devel

To download from the git repository I also needed git:
git (+ 25 packages)

The system had python3 already installed. But script test.sh needed python. Other packages I found I had to install to get to the compile stage are:
python-base (+ 2 packages)
ctags
lua-luarocks
clang7 (+21 packages)

Additional packages were also installed but unfortunately it is only after installation that I can be sure that the package contains, for example the include file, that is needed. Time taken: parts of 4 days (probably about 10 hours).

**Response:**

We are very sorry for the problems you met. The release package is improved and robust. The configure file can detect the environment and generate corresponding Makefile, thus saving lots of trouble. We believe the installation of the revised OpenArray is much easier as we have tested on many platforms.

[revised manuscript text omitted]

---

## Author Response (AR3)

Dear editor and reviewer,

First of all, we would like to express our sincere appreciation to your efforts installing OpenArray and valuable feedbacks. Your comments are highly insightful and enable us to substantially improve the quality of OpenArray and the user manual.

Below are our point-to-point responses to all comments.

**1 Responses to Referee #3**

User Manual
Section 2 Installation
(1) Make it clear in the software requirements list, item 2 that you need only one of MPICH or openmpi or Intel MPI compiler.
**Response:**
Thanks for your helpful suggestions. Item2 has been modified to clarify the requirement of MPI library.

In section 2 Installation, We changed the original sentence "2) Message Passing Interface (MPI) compilers (MPICH v3.2.1 or later; Openmpi v3.0.0 or later; Intel MPI compiler 2017 or later)." into "2) A Message Passing Interface (MPI) library, MPICH (version 3.2.1 or later) or Openmpi (version 3.0.0 or later) or Intel MPI compiler (version 2017 or later)."

Section 2.1 installation on Linux
(2) Make it clear that you are assuming a "bash" shell, and the "export" commands will need to be altered for other shell environments.
**Response:**
We appreciate your valuable suggestions. To make it clear, we added an example for different shell environments in the section 2 and section 4 (shown as below).

In section 2.1 installation on Linux, we added
"The way to set environment variables is different for different types of shell, so you may need to use 'setenv' or 'export' or other command, an example is listed below:
    setenv MPICC mpicc     (for csh or tcsh)
    export MPICC=mpicc (for sh, zsh or bash)

Below we are assuming the user is using sh, zsh, or bash.
"

In section 2.2 installation on Mac OS, we added:
"The way to set environment variables is different for different types of shell, so you may need to use 'setenv' or 'export' or other command, an example is listed below.
    setenv MPICC mpicc     (for csh or tcsh)
    export MPICC=mpicc (for sh, zsh or bash)

Below we are assuming the user is using sh, zsh, or bash.
"

In section 4.2.2 installation, we added:

"For sh, zsh, or bash:
    *export OPENARRAY_DIR=${HOME}/install*
    *export PNETCDF_DIR=${HOME}/install*

For csh or tcsh :
    *setenv OPENARRAY_DIR ${HOME}/install*
    *setenv PNETCDF_DIR ${HOME}/install*
"

(3) The default installation directory for PnetCDF is actually /usr/local which may be fine for many users. The default installation directory for OpenArray is actually /usr/local (checked with ./configure --help).

**Response:**

Thanks for your efforts installing OpenArray and your comments. In the manual of OpenArray, we assume to install PnetCDF and OpenArray in *${HOME}/install* since the default installation directory for PnetCDF and OpenArray (/usr/local) may cause permission issues for non-root users. To make it clearer, we revised the following sentences in the section 2.

In section 2.1 and 2.2, we added the following sentences: "Note: we are assuming *${HOME}/install* as the installation path of PnetCDF and OpenArray, since the default */usr/local* directory may cause permission issues for non-root users. However, the users can change it by specifying a different *PREFIX* value."

In section 2.1 item (b), we changed the original sentences "The default installation directory of PnetCDF is *${HOME}/install*" into "The default installation directory of PnetCDF is */usr/local* which might cause permission issues for non-root users. Thus we are assuming to install PnetCDF in *${HOME}/install*."

In section 2.1 item (c), we changed the original sentences "The default installation directory of OpenArray is *${HOME}/install*." into "Assuming the installation directory of OpenArray is *${HOME}/install*."

In section 2.2 item (c), we changed the original sentences "The installation directory of PnetCDF is *${HOME}/install*." into "The default installation directory of PnetCDF is */usr/local* which might cause permission issues for non-root users. Thus we are assuming to install PnetCDF in *${HOME}/install*."

In section 2.2 item (d), we changed the original sentences "The default installation directory of OpenArray is *${HOME}/install*." into "Assuming the installation directory of OpenArray is *${HOME}/install*."

(4) You should give example output from *./manual_main* (at least a few lines) for the user to compare to.

**Response:**

Thanks for your helpful suggestions. We added the example outputs from *./manual_main* and *./GOMO* in the manual shown as the following figures.

```
[huangxing@earth OpenArray-master]$ ./manual_main zeros =
         data type = float
         pos = -1
         is_pseudo = 0
         bitset = 111
         global_shape = [2, 2, 2]
         procs_shape = [1, 1, 1]
         bound_type = [0, 0, 0]
         stencil_type = 1
         stencil_width = 1
         lx = [2]
         ly = [2]
         lz = [2]
         clx = [0, 2]
         cly = [0, 2]
         clz = [0, 2]
[k = 0]
         j = 0                    j = 1
i = 0    0.000000000000000    0.000000000000000
i = 1    0.000000000000000    0.000000000000000
[k = 1]
         j = 0                    j = 1
i = 0    0.000000000000000    0.000000000000000
i = 1    0.000000000000000    0.000000000000000

ones =
         data type = float
         pos = -1
         is_pseudo = 0
         bitset = 111
         global_shape = [2, 2, 2]
         procs_shape = [1, 1, 1]
         bound_type = [0, 0, 0]
         stencil_type = 1
         stencil_width = 1
         lx = [2]
         ly = [2]
         lz = [2]
         clx = [0, 2]
         cly = [0, 2]
         clz = [0, 2]
[k = 0]
         j = 0                    j = 1
i = 0    1.000000000000000    1.000000000000000
i = 1    1.000000000000000    1.000000000000000
[k = 1]
         j = 0                    j = 1
i = 0    1.000000000000000    1.000000000000000
i = 1    1.000000000000000    1.000000000000000
```

Figure 1. The first few lines of the output from *./manual_main*.

```
[huangxing@earth bin]$ ./GOMO
start reading init variables...
nsbdy =           1 iint=           1
vamax =     5.0333135650657786E-002
* * *
time=       0.0020833;   iint=       1;   iext=       31;   iprint=       6
* * *
   vtot=       393292026336165.0625;   atot=   89167177231.1113892;   eaver=           -0.0006871
  taver=             8.3401626508;   saver=           35.0000000000;   tsalt=   13765220921765788.0000000
nsbdy =           1 iint=           2
vamax =     4.0942026145322240E-002
* * *
time=       0.0041667;   iint=       2;   iext=       31;   iprint=       6
* * *
   vtot=       393291977119409.3750;   atot=   89167177231.1113892;   eaver=           -0.0012391
  taver=             8.3401626465;   saver=           35.0000000000;   tsalt=   13765219199183138.0000000
nsbdy =           1 iint=           3
vamax =     4.7123633123959043E-002
* * *
time=       0.0062500;   iint=       3;   iext=       31;   iprint=       6
* * *
   vtot=       393291928442704.7500;   atot=   89167177231.1113892;   eaver=           -0.0017850
  taver=             8.3401626547;   saver=           35.0000000002;   tsalt=   13765217495563576.0000000
nsbdy =           1 iint=           4
vamax =     4.6187087118355202E-002
* * *
time=       0.0083333;   iint=       4;   iext=       31;   iprint=       6
* * *
   vtot=       393291827124303.8750;   atot=   89167177231.1113892;   eaver=           -0.0029213
  taver=             8.3401627246;   saver=           35.0000000001;   tsalt=   13765213949383962.0000000
nsbdy =           1 iint=           5
vamax =     4.3295450442905815E-002
* * *
time=       0.0104167;   iint=       5;   iext=       31;   iprint=       6
* * *
   vtot=       393291637375164.0000;   atot=   89167177231.1113892;   eaver=           -0.0050493
  taver=             8.3401626654;   saver=           35.0000000007;   tsalt=   13765207308421706.0000000
```

Figure 2. The first few lines of the output from *./GOMO*.

**2 Changes in the manuscript**

(1) In the acknowledgements section, we changed the sentence "Xiaomeng Huang is supported by a grant from the State's Key Project of Research and Development Plan (2016YFB0201100) and the National Natural Science Foundation of China (41776010)." into "Xiaomeng Huang is supported by a grant from the State's Key Project of Research and Development Plan (2016YFB0201100), the National Natural Science Foundation of China (41776010), and Center for High Performance Computing and System Simulation of Pilot National Laboratory for Marine Science and Technology (Qingdao)" (Lines 768~772)

(2) In the figure section, we modified the Fig.1 (shown as below) and changed the font in Fig.1~9 and Fig.12~13.

[revised manuscript text omitted]